# A comparison of humans and baboons suggests germline mutation rates do not track cell divisions

Felix L. Wu [ID][1,2]*, Alva I. Strand [ID][3¤a], Laura A. Cox[4,5], Carole Ober [ID][6], Jeffrey D. Wall[7], Priya Moorjani[3☯¤b], Molly Przeworski [ID][1,3☯]*

1 Department of Systems Biology, Columbia University, New York, New York, United States of America, 2 Integrated Program in Cellular, Molecular, and Biomedical Studies, Columbia University, New York, New York, United States of America, 3 Department of Biological Sciences, Columbia University, New York, New York, United States of America, 4 Center for Precision Medicine, Department of Internal Medicine, Section of Molecular Medicine, Wake Forest School of Medicine, Winston-Salem, North Carolina, United States of America, 5 Southwest National Primate Research Center, Texas Biomedical Research Institute, San Antonio, Texas, United States of America, 6 Department of Human Genetics, The University of Chicago, Chicago, Illinois, United States of America, 7 Institute for Human Genetics, Department of Epidemiology & Biostatistics, University of California, San Francisco, San Francisco, California, United States of America

☯ These authors contributed equally to this work.
¤a Current address: Department of Biology, Corix Plains Institute, Oklahoma Biological Survey, University of Oklahoma, Norman, Oklahoma, United States of America
¤b Current address: Department of Molecular and Cell Biology, Center for Computational Biology, University of California, Berkeley, Berkeley, California, United States of America
* flw2113@cumc.columbia.edu (FLW); mp3284@columbia.edu (MP)

**Data Availability Statement:** Pedigree and de novo mutation data are available within S2 Table, S3 Table, and S1 Data. Whole genome sequence data

## Abstract

In humans, most germline mutations are inherited from the father. This observation has been widely interpreted as reflecting the replication errors that accrue during spermatogenesis. If so, the male bias in mutation should be substantially lower in a closely related species with similar rates of spermatogonial stem cell divisions but a shorter mean age of reproduction. To test this hypothesis, we resequenced two 3–4 generation nuclear families (totaling 29 individuals) of olive baboons (*Papio anubis*), who reproduce at approximately 10 years of age on average, and analyzed the data in parallel with three 3-generation human pedigrees (26 individuals). We estimated a mutation rate per generation in baboons of $0.57\times10^{-8}$ per base pair, approximately half that of humans. Strikingly, however, the degree of male bias in germline mutations is approximately 4:1, similar to that of humans—indeed, a similar male bias is seen across mammals that reproduce months, years, or decades after birth. These results mirror the finding in humans that the male mutation bias is stable with parental ages and cast further doubt on the assumption that germline mutations track cell divisions. Our mutation rate estimates for baboons raise a further puzzle, suggesting a divergence time between apes and Old World monkeys of 65 million years, too old to be consistent with the fossil record; reconciling them now requires not only a slowdown of the mutation rate per generation in humans but also in baboons.

are available through the NCBI database of Genotypes and Phenotypes, (dbGaP study id phs000185, https://www.ncbi.nlm.nih.gov/gap/) and the NCBI Sequence Read Archive (BioProject accession number PRJNA433868, https://www.ncbi.nlm.nih.gov/bioproject/PRJNA433868); accession numbers are included in S1 Data. Data underlying the figure panels are available in S2 Data. Code for reproducing the figures and analyses is available at https://github.com/flw88/baboon_dnms.

**Funding:** This work was supported by National Institutes of Health/National Institute of General Medical Sciences (https://www.nigms.nih.gov) grant R01GM122975 to MP; a Burroughs Wellcome Fund Career at the Scientific Interface award (https://www.bwfund.org) and Alfred P. Sloan Foundation Fellowship (https://sloan.org) to PM; National Institutes of Health/Eunice Kennedy Shriver National Institute of Child Health and Human Development (http://www.nichd.nih.gov) grant R01HD21244 and National Institutes of Health / National Heart, Lung, and Blood Institute (https://www.nhlbi.nih.gov) grant R01HL085197 to CO; and National Institutes of Health/Office Of The Director (https://www.nih.govinstitutes-nih/nih-office-director) grant R24OD017859 to JDW and LAC. FLW was supported in part by National Institutes of Health/National Institute of General Medical Sciences (https://www.nigms.nih.gov) training grant T32GM008224. The funders had no role in study design, data collection and analysis, decision to publish, or preparation of the manuscript.

**Competing interests:** The authors have declared that no competing interests exist.

**Abbreviations:** AIC, Akaike information criterion; bp, base pair; CpG, 5′-cytosine-phosphate-guanine-3′; DNM, de novo mutation; FDR, false discovery rate; FNR, false negative rate; LR, Likelihood Ratio; MLE, maximum likelihood estimate; MRCA, most recent common ancestor; My, million years; SNP, single nucleotide polymorphism; SSC, spermatogonial stem cell.

## Introduction

Germline mutations are the source of heritable differences among individuals. They arise from accidental changes to the genome that occur in the development of a future parent, in the cell lineage from zygote to gamete (egg or sperm), either as errors in replication or due to damage that is improperly corrected or uncorrected by the next round of DNA replication. The number of de novo mutations (DNMs) in an offspring is thus an aggregate over the outcomes of mutational processes that play out from zygote to primordial germ cell and across the cell states of paternal and maternal gametogenesis.

Interestingly, the rate at which DNMs are introduced each generation varies substantially among species: As an illustration, across vertebrates surveyed to date, per base pair (bp) per-generation mutation rates span an order of magnitude [1]. Analyses of DNM and polymorphism data further indicate that in addition to the total rate, the proportion of each mutation type (the "mutation spectrum") also varies among mammalian species [2–4] and even across human populations [5–8].

The composite nature of the per-generation mutation rate allows for many possible explanations for these observations. At the cellular level, the damage rates to which species are exposed can change over time [9], as can repair and replication efficiencies. The cellular composition of the germline may also evolve—for example, the number of replications per generation. Shifts in any of these components can alter the overall mutation rate. These changes could occur by chance, by genetic drift, or due to natural selection on the mutation rate itself. Indeed, the mutation rate is not only the input to heritable differences but itself a phenotype, subject to genetic drift and selection [10,11].

The mutation rate may also evolve as a byproduct of changes in the life history of the species, e.g., the ages at which males and females typically reach puberty and reproduce. Because the life history of the species modulates the length of exposure of the germ cell lineage to distinct developmental stages, shifts in life history can lead to evolution in the per-generation mutation rate and spectra. A variant of this model has long been invoked to explain the observation that shorter-lived mammalian species (such as rodents) tend to have higher rates of neutral substitutions per unit time than longer-lived ones (such as primates) [12,13]. The "generation time effect" posits that species that are shorter-lived accrue more mutations per unit time because they undergo more cell divisions per unit time [12–16]. Although other correlated traits (such as body size, metabolic rate, and sperm competition) have been proposed to explain the observed dependence between reproductive span and mutation rates per unit time, in mammals, generation time remains the strongest predictor [13].

Evaluating these possible explanations requires comparative data on germline mutation from closely related species that share much of their developmental program but differ in some key features. In principle, such data are now straightforward to collect, by resequencing genomes from tissue samples (for practical reasons, often blood) of mother, father, and child and estimating the number of DNMs that occurred in that one generation. This approach has been applied to humans in numerous studies, confirming that most mutations are paternal in origin and revealing a strong, linear paternal age effect as well as a weaker maternal age effect on the number of DNMs [17–19]. Analogous data have also been collected in much smaller numbers of trios from other mammals, including mice [4], wolves [20], cattle [2], and primates (see Table 1).

Given the wealth of information for humans, comparisons to other primates may be particularly informative: This relatively closely related group of species varies markedly in their life histories, metabolic rates, and other potentially salient factors, and there exist well-documented differences in their per-year substitution rates [28,29]. Among the handful of apes and

**Table 1. Pedigree estimates of mutation rates and male-to-female DNM ratios in mammals.**

| Species | Common name | Primate lineage | Mean paternal age at conception | Sex-averaged mutation rate per generation, as reported ($\times 10^{-8}$) | Point estimate of α, ratio of paternal-to-maternal DNMs | Number of trios | References |
|---|---|---|---|---|---|---|---|
| *P. anubis* | Baboon | OWM | 10.27 | 0.57 | 4.50 | 12 | this study |
| *Homo sapiens* | Human | Great ape | 34.29 | 1.23 | 4.00 | 10 | this study |
| *H. sapiens* (Jónsson and colleagues) | Human | Great ape | 31.63 | 1.29[a] | 4.05 | 225 | [17] |
| *Pan troglodytes* (Besenbacher and colleagues)[b] | Chimpanzee | Great ape | 19.27 | 1.26[c] | 4.37 | 7 | [21,22] |
| *P. troglodytes* (Tatsumoto and colleagues) | Chimpanzee | Great ape | 24 | 1.48 | 3.08 | 1 | [23] |
| *Gorilla gorilla* | Gorilla | Great ape | 13.5 | 1.13[c] | 2.00 | 2 | [22] |
| *Pongo abelii* | Orangutan | Great ape | 31 | 1.66 | 4.13 | 1 | [22] |
| *Macaca mulatta* | Macaque | OWM | 7.5 | 0.58 | 3.21 | 14 | [24] |
| *Aotus nancymaae* | Owl Monkey | NWM | 5.55 | 0.81 | 2.09 | 14 | [25] |
| *Mus musculus* | Mouse | | 0.44 | 0.39 | 2.76 | 40 | [4] |
| *Bos taurus* | Cattle | | 5[d] | 1.2 | 2.53 | 13 | [2,26,27] |

Compilation of results from pedigree studies in mammalian species that measured sex-specific mutation rates.

[a]Includes single base pair indels.

[b]Includes the reanalysis of a pedigree from Venn and colleagues [21].

[c]Mean of reported rates per individual.

[d]Typical age at reproduction drawn from literature. All other ages are as reported in the relevant study.

DNM, de novo mutation; NWM, New World monkey; OWM, Old World monkey.

Old World monkeys (OWMs) from which direct estimates have been obtained, the estimated per-generation mutation rate varies over 2-fold. This interspecies variation has been interpreted largely in light of differences in life histories and cell division rates among species (e.g., [22,24,25]).

These interpretations rely on the assumption that mutations track cell divisions, as expected in various settings—most obviously, if mutations are due to errors in replication [16]. The number of mutations then increases with each DNA replication cycle, at a rate that depends on the fidelity of DNA replication in that cell type [16]. If, instead, mutations arise from damage that is inefficiently repaired relative to the length of the cell cycle, and the damage rate remains fairly constant over ontogenesis, then the accumulation of mutations may track time (age) rather than numbers of cell divisions [16]. In this case, differences among species may be more reflective of the damage rates that they experience and their typical ages at reproduction. Although the relative importance of different sources of mutagenesis remains unknown, in humans, 3 lines of evidence suggest that many germline mutations are not tracking cell divisions: (1) There is a maternal age effect on mutation, which likely reflects damage accumulating in oocytes [17,18]; (2) the spectrum of mutation types in males and females is similar (though not identical) [17,30,31], which one might not expect if mutations in oocytes arise from damage, but those that accrue during spermatogenesis are due to replication errors; and (3) the ratio of male-to-female mutations barely increases with parental ages, despite the fact that oocytogenesis is completed long before reproduction, whereas spermatogenesis is ongoing throughout life [32]. Given these observations from humans, it is unclear what one should expect when comparing them to other species.

Here, we aimed to compare mutation patterns in humans to those in olive baboons, an OWM with a younger reproductive age. This comparison is of interest because male baboons typically enter puberty (and hence spermatogenesis) around age 6 and have an average reproductive age of 10 years, with an estimated 33 spermatogonial stem cell (SSC) divisions per year, post-puberty [33,34]. In contrast, human males typically enter puberty at age 13 and reproduce at an average age of 32 years, with an estimated 23 SSC divisions per year [35–38]. Although there is considerable uncertainty about these numbers, they suggest that human sperm may be the product of >2-fold more cell divisions post-puberty relative to baboon sperm. This interspecies contrast thus sets up 2 predictions. First, if mutations are replicative in origin and rates per cell division have not evolved, we expect fewer mutations to arise from spermatogenesis in baboons compared with humans. In other words, all else being equal, we would expect the fraction of mutations contributed by baboon fathers to be lower than that of human fathers. Second, there should be a stronger paternal age effect in baboons, in keeping with their faster rate of SSC division.

A number of technical difficulties stand in the way of this comparison. Germline mutations are rare (on the order of 100 in a 6-Gb genome) [39]. Reliable estimates are therefore exquisitely sensitive to false negative and positive rates, which, in turn, depend on sequencing coverage, the reference genome quality, and the variant-calling pipeline. Within humans, different studies have arrived at discrepant estimates of parental age effects, mutation spectra, and mutation rates at a given paternal age [40]; small but significant differences are seen even between estimates based on 2- versus 3-generation families within a single study [32]. These observations strongly suggest that comparisons among species are likely to be confounded by the choice of analysis pipeline. To try to minimize these technical biases, we resequenced 3-generation families, calling putative DNMs in the second generation and then verifying transmission and phasing mutations with the third; moreover, we applied the same variant-calling pipeline to the 2 species, in orthologous regions of the genome.

## Results

### Identifying DNMs from pedigrees in parallel in the 2 species

In total, we resequenced the genomes of 26 humans from three 3-generation pedigrees and 29 olive baboons from two 3-generation pedigrees, to mean depths of 24X and 45X, respectively (Fig 1). After mapping the reads of each individual to its species' reference genome, we called variants using the Genome Analysis Toolkit (GATK) following standard quality control practices [41]. We identified single-nucleotide polymorphisms (SNPs) at which the $F_1$ child was heterozygous, whereas both $P_0$ parents were called homozygous for the reference allele, thus constituting a Mendelian violation. We filtered these variants on a variety of quality statistics and sequence metrics (see Methods) using the same criteria in both species, thereby identifying 1,055 putative autosomal DNMs in the 10 focal human $F_1$ individuals and 510 DNMs in the 12 focal baboon $F_1$ individuals (Fig 1). We additionally flagged putative DNMs occurring in clusters, which we defined as DNMs arising fewer than 100 bp away from a neighboring DNM. Analyzing transmission patterns and validation results from Sanger sequencing, we concluded that clusters containing more than 2 variants were likely artifacts and thus discarded them in subsequent analyses (see Methods, S1 Fig). After this filtering step, we retained a total of 980 human and 475 baboon putative DNMs.

By applying the same filtering pipeline to both species, we aimed to create a reliable comparison of their mutational properties. Furthermore, we identified regions of the genome that are orthologous between humans and baboons and in which we were highly powered to detect DNMs in all individuals based on sequencing depth (see Methods). After restricting our

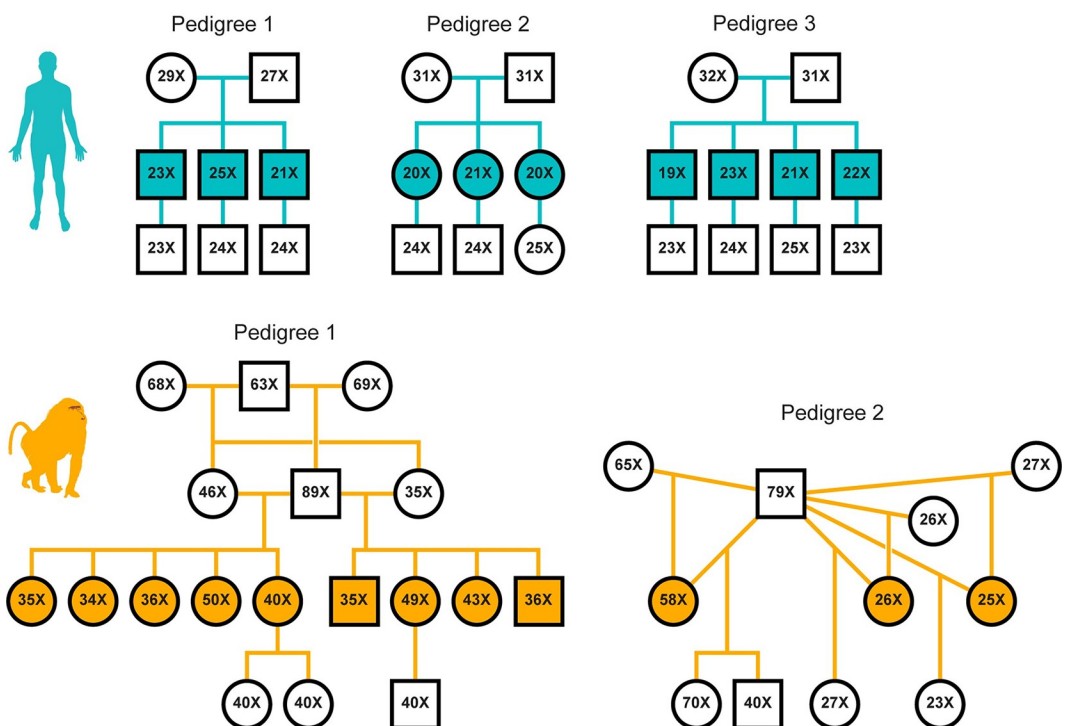

**Fig 1. Pedigrees and mean depth of coverage of the sequenced individuals.** Relationships among individuals in 3 human families (top row, teal) and 2 extended baboon pedigrees (bottom row, orange) are shown. Squares and circles represent males and females, respectively, and are annotated for the mean depth of sequencing coverage of the individual. Filled shapes indicate the focal $F_1$ individuals in which mutations were called. Sample identifiers and birth dates for all individuals are specified in S2 Table and S3 Table.

analysis to these orthologous regions, which spanned a combined total of approximately 1.6 Gb, 505 putative human DNMs and 248 baboon DNMs remained. Rates in the orthologous regions are only slightly lower than those estimated when including nonorthologous regions (i.e., using the entire callable genome for which the mean across trios is approximately 2.5 Gb in human and approximately 2.3 Gb in baboon; see "Estimating sex-specific germline mutation rates and age effects").

Even when relatively infrequent, spurious DNMs can have a non-negligible impact on the mutation spectrum and the estimated fraction of DNMs that are of paternal as opposed to maternal origin (bringing it closer to 1:1 than it is truly). We therefore ran a series of analyses to estimate false negative rates (FNRs) and false discovery rates (FDRs). First, for each trio, we generated a set of 20,000 simulated DNMs in such a way as to create Mendelian violations that mimic DNMs, then ran this set of simulated DNMs through our filtering pipeline and assessed the rate at which we failed to detect the mutation in the proband. This approach yielded median FNR estimates across trios of 10.22% (range: 9.11%–18.96%) in humans and 9.96% (range: 9.11%–11.65%) in baboons (S2A Fig). We note that, similarly to other studies, our approach to estimate error rates does not take into account read mapping errors. In principle, these could generate both false negatives and false positives; in practice, we suspect this limitation leads the resulting estimates of mutation rate to be slightly underestimated (see Methods).

To measure FDRs (i.e., the proportion of spurious mutations in our putative DNM call sets), we relied on the fact that sequencing errors called as DNMs in an $F_1$ individual are unlikely to be observed in its $F_2$ offspring, whereas genuine DNMs should be observed approximately half the time. In practice, we obtained FDR estimates using a maximum likelihood

based model (see Methods), which accounts for the proportion of DNMs called in both the $F_1$ and $F_2$ as well as the probability of observing transmission events in the first place (which is affected by the variant-calling pipeline and the sequencing coverage of the specific $F_2$). We estimated FDRs separately for each trio, with median values of 21% in humans and 18% in baboons (S2B Fig). To verify our transmission-based approach to FDR estimation, we Sanger-sequenced putative DNMs called in 1 human individual (both inside and outside of the orthologous regions; see S1 Table and S3 Fig). Using our transmission-based approach, we had obtained a maximum likelihood estimate of the FDR of 0% for this trio, with a 95% CI of 0%–21%; consistent with this estimate, we found only 1 false positive out of the 24 DNMs that we were able to check by Sanger sequencing ($p$-value = 0.35, by a Fisher's exact test, S4 Fig). In what follows, we used the transmission-based FDR for each trio.

## Estimating sex-specific germline mutation rates and age effects

In order to assess maternal and paternal age effects on mutation, we determined the parent-of-origin of the putative DNMs using a combination of read tracing and pedigree-based phasing (see Methods). In humans, we were able to phase a total of 162 DNMs by the first approach and 216 by the second, with highly concordant parental assignments for mutations that could be phased by both approaches (64 of the 65 DNMs). We then combined the phasing information with our estimated FDR and FNR values to infer the total number of maternally or paternally derived DNMs for each $F_1$ (scaled up to an autosomal haploid genome size of 2.881 Gb, based on hs37d5). Performing a Poisson regression of the maternal (paternal) human DNM counts against the mother's (father's) age at conception, we found a significant ($p$-value < 0.001) paternal age effect of 1.04 DNMs per year (95% CI 0.41–1.66), as well as a significant ($p$-value < 0.001) maternal age effect of 0.81 DNMs per year (95% CI 0.50–1.12).

Our estimate of the paternal slope, which is based on our sample of only 10 human trios (Fig 2A), is consistent with the one reported by Gao and colleagues, who relied on a sample of 225 3-generation trios (Likelihood Ratio [LR] test $p$-value = 0.34; matching the genome size surveyed) [17,32]. Our maternal slope, however, is slightly higher (LR test $p$-value = 0.014). A possible explanation is that our sample includes 2 trios with mothers over 40. Indeed, Gao and colleagues report a better fit of an exponential than linear model, driven by observations for mothers over 40, and likewise, such a model fits our data slightly better than their linear model (Akaike information criterion [AIC]; ΔAIC = −3.92). In turn, our sex-averaged germline mutation rate is $1.23×10^{-8}$ (95% CI $1.14–1.32×10^{-8}$) per bp, in agreement with previous estimates of the human mutation rate [17,30,32,42,43], for paternal ages of 32.0 years and maternal ages of 28.2 years (the mean generation times in a previous DNM study that sampled over 1,500 trios [17]). Together, these comparisons to larger data sets in humans suggest that our DNM filtering and discovery pipelines are reliable.

We therefore proceeded to estimate the same parameters in our set of baboon DNMs by the same approach. As before, we inferred sex-specific DNM counts in each $F_1$ by 2 phasing approaches and adjusted counts by estimated FDR and FNR. Given the higher diversity in baboons, we were able to phase a higher proportion of mutations by read-back phasing (126/248 = 51%), and given the lower number of third-generation individuals, we could phase only 56 by transmission; all but one of the 30 DNMs phased by both approaches were in agreement. For each sex separately, we performed a Poisson regression of mutation count estimates against the age at conception of the parent (Fig 2B). We found a paternal age effect of 0.15 DNMs per year (95% CI −0.97 to 1.32), which was not significant at the 5%-level ($p$-value = 0.79), possibly because of the narrow range of paternal ages in the study. Intriguingly, we obtained a significant maternal age effect of 0.55 DNMs per year (95% CI 0.03–1.02; $p$-value =

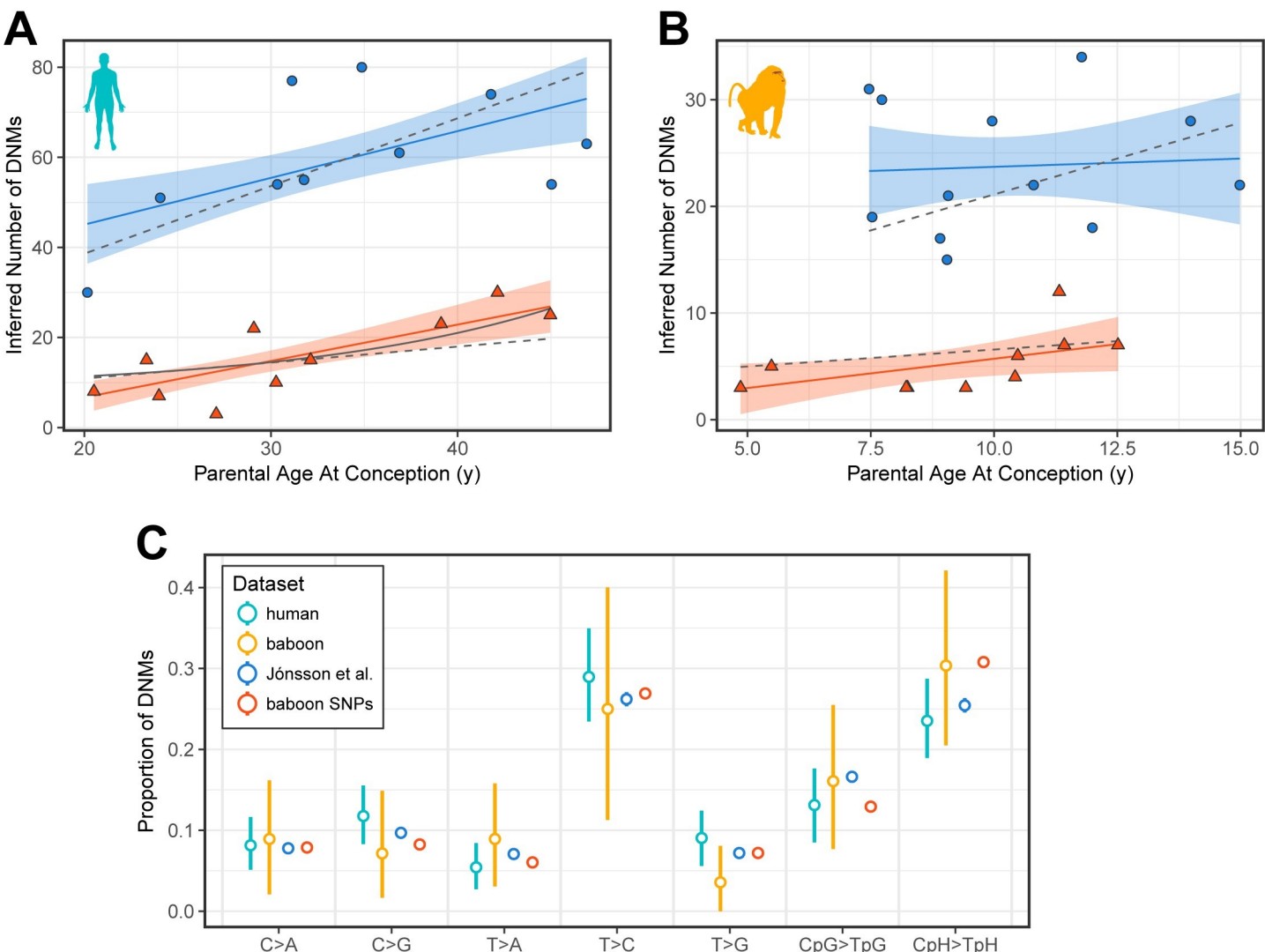

**Fig 2. Dependence of mutation count on sex and age.** Each human (in A) and baboon (in B) $F_1$ individual is represented by a blue circle and a red triangle, which indicate how many DNMs were estimated to have occurred on the paternal or maternal genomes, respectively. Maternal points for 2 baboons were omitted because of a lack of information about the age of the mothers. Solid blue and red lines indicate the best fit obtained from a Poisson regression for each sex, respectively. Blue- and red-shaded regions denote the 95% CIs associated with the regression coefficients. Gray dashed lines represent the sex-specific fits from Gao and colleagues [32], scaled to account for differences in genome size in [32] versus this study. In (A), the solid gray line denotes the exponential maternal age effect fit from Gao and colleagues. We did not find a significant difference between the parental slopes of the 2 species (LR test $p$-value = 0.19 in fathers, 0.37 in mothers). (C) Human and baboon mutation spectra, in our sequenced cohorts and previously published datasets. Spectra are shown for all transmitted DNMs in humans (teal); all transmitted baboon DNMs (orange); DNMs called in a large set of Icelandic 3-generation pedigrees [17] (blue); and low-frequency SNPs (doubleton and tripleton alleles) identified in a sample of distantly related baboons (red). Each point denotes the relative proportion of mutations that belong to 1 of 7 types of mutations, as indicated on the x-axis; reverse complement mutation types were collapsed together into a single type (e.g., the left-most column, C>A, includes G>T mutations). Transitions at cytosine sites were separated into those that occurred at a CpG (CpG>TpG) or a non-CpG (CpH>TpH) site. For all datasets, only DNMs and SNPs located within regions of the genome determined to be orthologous between humans and baboons were included. Vertical lines represent 95% CIs from bootstrap resampling of 50 cM blocks. CIs on the 2 reference datasets (blue and red) were constructed by bootstrap resampling of mutations but are small and hidden by the points. Underlying data for this figure can be found in S2 Data. cM, centimorgan; CpG, 5′-cytosine-phosphate-guanine-3′; DNM, de novo mutation; LR, likelihood ratio; SNP, single-nucleotide polymorphism.

0.043); however, the $p$-value for the maternal age effect is strongly dependent on the inclusion of one somewhat outlier point. To obtain a per-generation mutation rate estimate for baboons, we assumed that the mean ages of wild baboon fathers and mothers are 10.7 years and 10.2 years, respectively (Jenny Tung, personal communication), which yields $0.57 \times 10^{-8}$ (95% CI $0.51$–$0.64 \times 10^{-8}$) per bp, approximately 46% of what we had inferred for humans.

Reanalyzing the data considering all callable regions, not only orthologous ones, yielded a similar ratio between the 2 species, with a per-generation mutation rate of $1.33 \times 10^{-8}$ per bp in humans (95% CI $1.23–1.43 \times 10^{-8}$) and $0.62 \times 10^{-8}$ (95% CI $0.55–0.69 \times 10^{-8}$) in baboons (S5A Fig). In both species, this slight increase appears to be driven by higher mutation rates in the repetitive, nonorthologous regions of the genome relative to nonrepetitive, nonorthologous regions (S5B Fig; Poisson rate ratio test $p$-value < 0.001 for both species). Whether the increase in repetitive regions represents a true increase in the mutation rate or a higher error rate remains to be determined.

In summary, our findings in baboons suggest that the per-generation mutation rate in OWMs is approximately half that of great apes, in which direct estimates range from $1.13 \times 10^{-8}$ per bp in gorillas to $1.68 \times 10^{-8}$ per bp in orangutans [22]. Existing point estimates for OWM species are generally lower than those of apes, varying from a point estimate of $0.58 \times 10^{-8}$ per bp in rhesus macaques to $0.94 \times 10^{-8}$ per bp in green monkeys [24,44]. These numbers are based on small numbers of trios and are thus imprecise (see Table 1). More critically, differences in how studies chose to filter DNMs and estimate error rates currently hinder reliable conclusions about how mutation rates differ across species. We sought to minimize these issues by identifying DNMs in a parallel manner across 2 species in the same regions of the genome, thus strengthening the case for a substantially lower mutation rate per generation in OWMs compared with apes.

## The mutation spectra in baboons and humans

To understand how the distribution of single bp mutation types (i.e., the mutation "spectrum") compares between humans and baboons, we analyzed our DNMs alongside existing mutation and polymorphism datasets (Fig 2C). We focused our analysis on the subset of DNMs that were observed at least once in the $F_2$ generation as well as the $F_1$ (total of 216 in humans and 56 in baboons), given that these are highly unlikely to be spurious. We classified each DNM into 7 disjoint mutation classes: C>A, C>G, T>A, T>C, T>G, CpG>TpG, and CpH>TpH; the latter 2 classes denote mutation types occurring within and outside of a 5′-cytosine-phosphate-guanine-3′ (CpG) context, respectively. In humans, the mutation spectrum in our trios was not significantly different from proportions obtained from a set of 8,895 human DNMs identified from 225 3-generation pedigrees by Jónsson and colleagues [17]. Similarly, in baboons, we did not observe any significant differences in the mutation spectra when comparing our de novo calls against low-frequency SNPs, which should closely reflect the DNM spectrum (forward variable selection $p$-value > 0.05 for all comparisons). Repeating this analysis with all mutations, including those that were not observed as transmitted to an $F_2$, yielded largely similar results (S6 Fig). These results about the mutation spectrum provide further evidence for the reliability of our calling pipeline.

Comparing the larger human DNM dataset from Jónsson and colleagues and the baboon SNP dataset, there are significant differences in the proportions of 4 mutation types (forward variable selection $p$-value < $10^{-4}$ for C>G, T>A, CpG>TpG, and CpH>TpH), consistent with differences in mutation spectra found among other primate species in analyses of polymorphism and divergence [3,6]. Using our small de novo datasets, however, we lack power to detect such subtle differences (forward variable selection $p$-value > 0.05 for all comparisons).

## The strength of male germline mutation bias is similar in baboons and humans

We considered sex-specific mutation rates in the 2 species in light of what we would predict under a simplified model in which all germline mutations are replicative in origin. If we further assume that human and baboon ova are the product of the same numbers of cell divisions

and the same per-cell division mutation rates, then we would expect similar per-generation mutation rates in females of the 2 species. Instead, the mutation rate is $0.23 \times 10^{-8}$ (95% CI 0.20–$0.27 \times 10^{-8}$) at age 28.2 in humans and $0.11 \times 10^{-8}$ (95% CI 0.082–$0.14 \times 10^{-8}$) at age 10.2 in baboons, a 2-fold difference. Given recent evidence in humans that a substantial proportion of female mutations arise from damage to oocytes [17,19,32], the elevated rates in humans could result from the substantially older age of human oocytes compared with those of baboons. Indeed, we fail to reject a model in which the mutation rate difference between human and baboon mothers simply reflects their typical mean ages at reproduction (LR test $p$-value = 0.26).

In fathers, SSC division rates have been estimated to be 23 cell divisions per year in humans versus 33 in baboons [34,37]. Thus, if the mutation rate per SSC division is the same in the 2 species, we expect a 1.5-fold stronger paternal age effect in baboons compared with humans. Yet the baboon data are highly unlikely under a paternal age effect of that magnitude or greater (LR test $p$-value = $2.8 \times 10^{-3}$). If anything, it appears as if paternal mutations are accruing more slowly per year in baboons than in humans (LR test $p$-value = 0.046), even after excluding CpG transitions, which are thought to stem mostly from spontaneous deamination [15] (LR test $p$-value = 0.012). The data further suggest that baboon males may have accumulated fewer DNMs by puberty than human males (LR test $p$-value on the intercept = 0.047) (S7 Fig) and, given limited data, are consistent with a model in which the higher intercept simply reflects the 2.4-fold older age of human males at puberty (LR test $p$-value = 0.12). Thus, as far as we can tell with a small number of baboon trios, parental age effects seem to reflect ages rather than what is known about cell division rates.

Next, to compare the male-to-female mutation ratio, $\alpha$, in the 2 species, we focused our analysis on the set of DNMs called in the $F_1$ that were observed in the $F_2$ generation. In our human samples, we estimated that 80% (95% CI 74%–85%) of the DNMs were paternal in origin—i.e., an $\alpha$ of approximately 4.0—which is consistent with previous estimates for the same ratio of paternal-to-maternal ages [17,19]. In baboons, we obtained a paternal fraction of 82% (95% CI 70%–93%), or an $\alpha$ of 4.5, which is not significantly different from the value in humans ($\chi^2$ test $p$-value = 0.91) (Fig 3A). Under the assumptions about cell division number and rates outlined previously, we would expect the ratio of male-to-female mutations to be roughly 2.2-fold larger in humans than in baboons (see Methods). We readily reject this model (LR test $p$-value = $3.84 \times 10^{-8}$), even after removing transitions at CpG sites (LR test $p$-value = $2.30 \times 10^{-7}$). Excluding these CpG transitions also yields similar $\alpha$ estimates: 4.75 in baboons (paternal fraction 83%, 95% CI 72%–93%) and 3.9 in humans (paternal fraction 80%, 95% CI 73%–85%). Thus, although assumptions about the number of cell divisions in humans and baboons are tenuous, it is clear that the degree of male bias in mutation is surprisingly similar in humans and baboons.

To explore this observation further, we compared our estimates of $\alpha$ to what has been reported from direct pedigree analyses of other mammals, whose paternal generation times vary markedly, from months to years to decades. Although these studies are based on scant data, the point estimates of $\alpha$ are remarkably similar in all 7 species, with almost no increase with paternal age at reproduction (Fig 3B). This observation echoes what is seen within humans, where despite the increase in the number of SSC divisions with paternal age, the male bias is already 3:1 by puberty and is relatively stable with parental ages [32].

## Discussion

### Implications for the genesis of mutation

We set out to test whether differences in baboon and human mutation rates are readily explained by their life histories. As a starting point, we considered a simple model in which

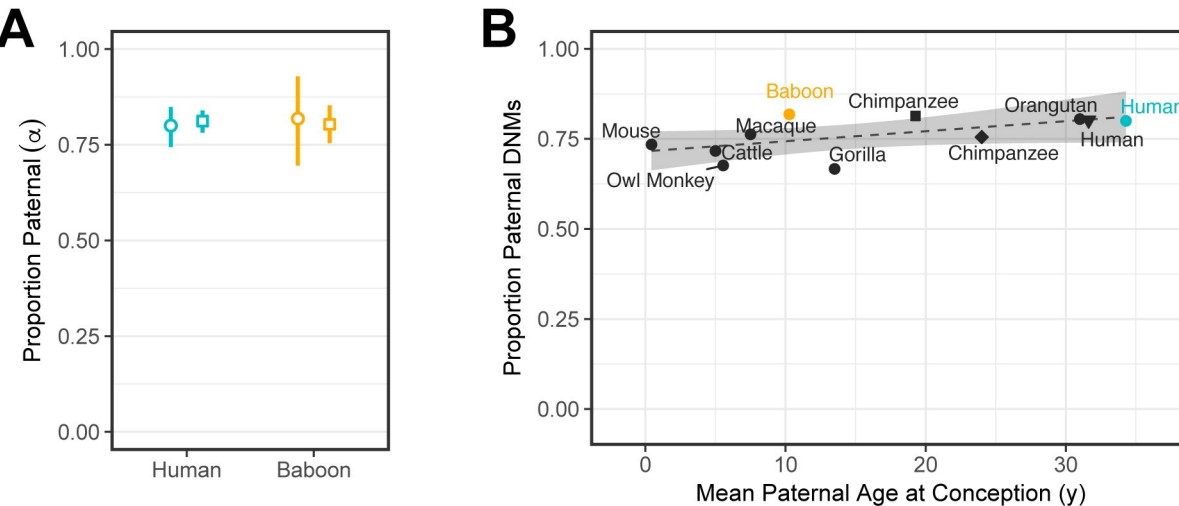

**Fig 3. Estimates of the sex bias in mutation across mammals.** (A) Ratio of paternal-to-maternal mutations, $\alpha$, among phased DNMs. Circles indicate the fraction of DNMs assigned to the paternal genome among all transmitted DNMs (i.e., DNMs observed in an F2 offspring); in our samples, the mean ages in humans were 34.3 years for males and 31.3 years for females and in baboons, 10.3 years and 9.2 years, respectively (so the ratio of male-to-female generation time was about 1.1 in both). Vertical lines indicate 95% CIs, which were determined by bootstrap resampling of 50 cM blocks. Squares denote the ratio of paternal-to-maternal mutations estimated for typical parental ages in the species, assuming ages 32.0 and 28.2 years for human males and females, respectively, and 10.7 and 10.2 years for baboon males and females. Vertical lines denote the 95% CIs of the age effect and intercept estimates, obtained assuming normally distributed errors. (B) The ratio of paternal-to-maternal DNMs estimated from pedigrees across 9 mammalian species as a function of the mean male generation times in the study (for references, see Table 1). The square and diamond points denote separate chimpanzee estimates from Besenbacher and colleagues and Tatsumoto and colleagues, respectively [22,23]. The black triangle denotes the estimate for DNMs phased by transmission in 225 3-generation human pedigrees from Jónsson and colleagues [17]. Human and baboon estimates reported in this study are colored teal and orange, respectively. Fitting a linear regression (dashed line) yields an estimated slope of only $2.8 \times 10^{-3}$ (95% CI $-9.2 \times 10^{-4}$ to $6.5 \times 10^{-3}$, p-value = 0.12). To avoid duplicates, values from one chimpanzee (Tatsumoto and colleagues [23]) and one human (Jónsson and colleagues [17]) study were not included in the regression. The shaded area denotes the 95% CI of the slope and intercept. Underlying data for this figure can be found in S2 Data. cM, centimorgan; DNM, de novo mutation; SNP, single-nucleotide polymorphism.

mutations are proportional to cell divisions, females have the same number of cell divisions in the 2 species, and per-cell division mutation rates over ontogenesis are the same in the 2 species. Under these assumptions, we would expect a stronger paternal age effect in baboons and a higher male mutation bias in humans. Neither of these expectations are met: The paternal age effect is not discernably stronger in baboons, and the male bias is similar—as it is in other mammals for which direct pedigree estimates of sex-specific rates exist (Fig 3B).

A likely possibility is that some of our assumptions are wrong. In particular, a recent review argued that germline mutations are replicative in origin and hence track cell divisions but that the rate of SSC divisions is much lower than previously believed [45]. Although plausible [45,46], this explanation alone would not explain our findings: If mutations were due to replication errors and if rates of SSC divisions were very low, then without making further assumptions, we would expect human and baboon paternal mutation rates per generation to be highly similar, when they are not. In turn, the approximately 2-fold lower mutation rate observed in baboon compared with human females could be explained if there are fewer rounds of DNA replication in baboons than humans (or greater replication fidelity). Thus, individual observations can be explained under a replicative model by invoking specific parameters.

Taken together with other studies, however, our findings add to a growing set of observations that do not readily fit a model in which most germline mutations track cell divisions, including that (1) in humans and in baboons, there is a maternal age effect on mutations, which contributes a substantial proportion of maternal mutations, despite the absence of cell divisions after the birth of the future mother [17,19,32; this paper]; (2) the male mutation bias in humans

is already approximately 3:1 by puberty, when germ cells from the 2 sexes are thought to have experienced similar numbers of cell divisions by then [32]; (3) in humans, the male mutation bias barely increases with parental age, and even less so once CpG transitions are excluded, despite ongoing SSC divisions [32]; and (4) the sex bias in mutation rates is roughly similar across mammals [this paper; 4]. Observation 1 indicates that a non-negligible fraction of mutations in females are nonreplicative. Observations 2–4 could be explained by replication errors if we assume that a number of current beliefs about spermatogenesis are incorrect: namely, that there are more (or more mutagenic) cell divisions in males than females before puberty and that there are fewer (or less mutagenic) cell divisions in males after puberty. Even if both conditions hold, all the parameters would still need to conveniently cancel out both within humans and across mammals to generate an apparent dependence on absolute time rather than on cell divisions [16] and a relative stability of the male bias in mutation.

An alternative is that germline mutations are predominantly due to damage in both sexes. Accounting for all 4 observations would require damage to accrue at a relatively fixed rate across mammals, but at a somewhat higher rate in males than in females, and be inefficiently repaired relative to the cell cycle length [16]. In principle, this hypothesis could then explain the relative stability of the male mutation bias with parental ages in humans, the similarity of the male mutation bias across mammals, and the similarity of the mutation spectrum in the 2 sexes. It would also explain why primate mutation rates per generation appear to roughly follow typical ages of reproduction [this paper; 25]. However, it too requires a number of assumptions, and these remain to be tested (e.g., [47]).

## Comparing contemporary mutation rates and substitution rates

Studies of divergence in primates clearly demonstrate that neutral substitution rates vary substantially across the phylogeny [29]. Notably, the olive baboon has accrued 35% more substitutions along its lineage as compared with humans since their common ancestor (Fig 4A). If neutral, mutations are expected to fix at the rate at which they arise [48]. Thus, we would expect the mutation rate per unit time in baboons to be substantially higher than that in humans. To evaluate this hypothesis, we converted our de novo germline mutation rates to yearly rates using a sex-averaged model that accounts for sex-specific relationships of mutation rates to age and sex-specific life history traits [42]. This yielded yearly mutation rates of $5.49 \times 10^{-10}$ in baboons, 35% (95% CI 18%–51%) larger than the rate of $4.08 \times 10^{-10}$ in humans (Fig 4B). Thus, the ratio of present-day yearly mutation rates appears to be quite consistent with the observed substitution rate ratio in these 2 species.

Because biased gene conversion on mutation acts like selection and influences the substitution process, we further broke up substitutions by type, depending on whether the fixation probability was increased, decreased, or unaffected by biased gene conversion; our findings are as expected, with mutations favored (or disfavored) by biased gene conversion showing slightly higher (or lower) substitution rates relative to mutation rates (Fig 4C). Although imprecisely estimated, mutation types not subject to biased gene conversion (Strong [G/C] > Strong and Weak [A/T] > Weak) show good agreement between mutation and substitution rates.

If mutations are neutral and accumulate at a fixed rate, we can relate divergence levels to mutation rates in order to estimate the mean time to the most recent common ancestor (MRCA), in this case of OWMs and great apes. Dividing the human neutral substitution rate by the yearly mutation rate yields a time of 64 million years (My), whereas the same calculation using rates estimated in baboons yields a divergence time of 65 My. Yet evidence from the fossil record dates the OWM–great ape population split time to at most 35 My [49,50]. Although divergence-based estimates are for the mean time to the MRCA rather than the split time,

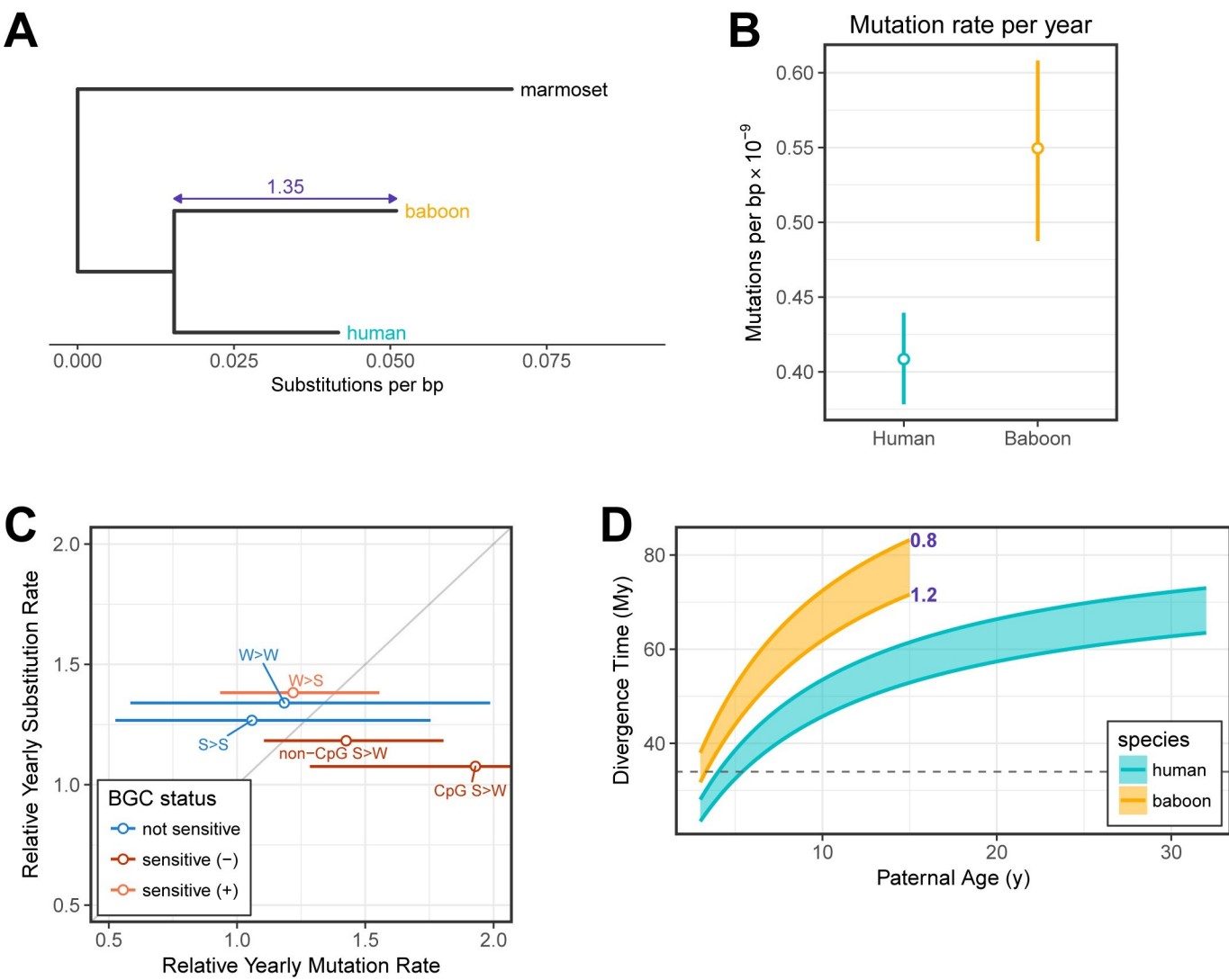

**Fig 4. Implications of present-day human and baboon mutation rates for the evolution of the yearly mutation rate.** (A) Phylogenetic relationship between humans and baboons with a marmoset (New World monkey) outgroup. Branch lengths denote the autosomal substitution rate per bp since the OWM–marmoset split as measured using data from [29] for all mutation types at putatively neutral regions of the genome. The relative branch length difference between baboon and human lineages is indicated in purple. (B) Sex-averaged mutation rates per year. Mutation rates were based on fitted values for typical generation times (i.e., assuming 32.0 and 28.2 years in human males and females, respectively, and 10.7 and 10.2 years in baboon males and females) and turned into a sex-averaged per-year mutation rate following [42]. Vertical lines indicate the span covered by the 95% CIs of the intercept and slope of the age effect regressions. (C) The ratio of yearly mutation and substitution rates in baboon relative to human, as estimated for the different possible types involving combinations of strong (S: G/C) and weak (W: A/T) bp. Each point denotes a different type, and strong-to-weak (S>W) types were separated into those that occurred at a CpG or a non-CpG site. Points are colored according to whether GC-biased gene conversion is expected to favor (light red), disfavor (dark red), or have no effect (blue) on the mutation type. Horizontal lines denote 95% CIs on the mutation rate ratio computed by resampling of 50 cM blocks. The upper CI for CpG S>W extends out of frame to 2.8. Point estimates for the substitution rates in baboons and humans were taken from [29]. The identity line is drawn in gray for reference. (D) Predicted divergence times of humans and OWMs as a function of parental ages. Divergence times were predicted using mutation and autosomal substitution rates measured in humans (teal) and baboons (orange), across a span of plausible past generation times. Each point within the shaded areas represents the divergence time calculated at a particular paternal generation time (x-axis) and paternal-to-maternal generation time ratio (ranging from 0.8 to 1.2) as indicated in purple. The dashed gray line indicates a plausible upper bound for the split time inferred from the fossil record [49,50]. Underlying data for this figure can be found in S2 Data. BGC, biased gene conversion; bp, base pair; cM centimorgan; CpG, 5′-cytosine-phosphate-guanine-3′; OWM, Old World monkey.

these two are expected to be very similar for species so diverged (in units of $N_e$ generations, where $N_e$ is the effective population size) [51]. Thus, the number of substitutions suggests a split time that is implausibly old.

A possible explanation is that yearly mutation rates in both human and baboon lineages have slowed towards the present because of changes in life history traits alone [14,28,42]. We explored this hypothesis by examining the effect of historical generation time on the inferred mean time to the common ancestor (Fig 4D). We varied paternal generation times ranging from a lower bound of 3 years (the average age of reproduction of various New World monkey species; [51]) in both species to upper bounds of 32 years in humans and 15 years in baboons; we further allowed the male-to-female generation time ratio to vary from 0.8 to 1.2 [42]. We used the age effect point estimates published by Gao and colleagues [32] as the age effect parameters of humans in our model, reasoning that—being drawn from a larger sample—these estimates would be more precise in humans. Given that we had too few baboon trios to estimate parameters precisely, we assumed that the strength of the parental age effects in baboons are similar to that of humans and used the estimates of Gao and colleagues to model mutation accumulation in baboons as well, despite tentative evidence that the baboon paternal age effect may be weaker (S7 Fig). Our analysis suggested that, in both species, implausibly low generation times of approximately 3–5 years would be required to yield divergence times that are more in line with fossil-based estimates and even then, barely so. Instead using our own estimates for the age effect parameters in baboon males and females leads to the same conclusion (S8 Fig). Thus, our results extend the puzzle first pointed out by Scally and Durbin [53] of an apparent disconnect between the evolutionary times suggested by phylogenetic and pedigree data in humans. Reconciling them now requires not only a slowdown of the mutation rate per generation in humans but also in baboons.

A parallel slowdown in both lineages seems highly unlikely if mutations are replicative in origin, given that changes in life history cannot plausibly explain the magnitude of the effect. In principle, one might imagine that germline mutation rates in the 2 species are shaped by the same exogenous mutagen and that a shift occurring after their split affected rates similarly in both lineages, leading to a parallel slowdown. If so, the change in damage rate could not have affected the ratio of male-to-female mutations much, because $\alpha$ appears to be similar in both species. To evaluate this possibility, it will be important to obtain comparable estimates from more species, in particular, an outgroup to OWM and apes, such as a New World monkey. An alternative is that the OWM fossil record has been misinterpreted to suggest a more recent split time of OWMs and apes than is truly the case.

More generally, although we have argued above that germline mutation patterns in humans and baboons are more easily explained if they are primarily due to damage, such a hypothesis does not provide an immediate explanation for why per-year mutation and substitution rates vary across mammalian species or tend to be higher in shorter-lived ones [12,14–16,54]. One possibility is that rates of damage co-vary somewhat with life history traits [55], with a tendency towards higher damage rates in shorter-lived species. In that regard, it would be interesting to characterize substitution rates in sets of species that differ in their environmental exposures and metabolic rates and examine differences in their mutation spectrum in light of known mutagens (e.g., [9]).

## Methods

### Ethics statement

This study was approved by the Institutional Review Board at the University of Chicago (protocol 8073) and the Institutional Review Board at Columbia University (protocol AAAN6358). All individuals provided informed, written consent to the use of their DNA sequences.

### Human DNA samples and whole genome sequencing

We resequenced the genomes of 26 Hutterite individuals [56,57]. The DNA samples were derived from either whole blood (25 individuals) or saliva (1 individual). We performed 100

bp paired-end sequencing of the PCR-free libraries, one per individual, to an average of 24X coverage on the Illumina HiSeq 2000 short-read sequencing platform at the University of Chicago.

Our cohort consisted of 3 multisibling families, containing 10 distinct parent–child trios (Fig 1). Throughout, the first generation is referred to as $P_0$, and the focal generation $F_1$. We also sequenced one offspring of each of the $F_1$ individuals, referred to as $F_2$, which allowed us to assess transmission of DNMs to a third generation. Ages of both $P_0$ parents at the birth of the $F_1$ offspring were recorded for all 10 trios (see S2 Table).

### Baboon DNA samples and whole genome sequencing

Our sample of olive baboon (*P. anubis)* individuals originated from the captive colony housed at The Southwest National Primate Research Center (SNPRC), which was established in the 1960s with founder animals from southern Kenya. We initially sequenced a single sire, 2 dams, and their 9 $F_1$ offspring (Pedigree 1 in Fig 1), obtaining baboon DNA samples from the SNPRC, which were previously extracted from archived liver or buffy coat samples. A paired-end 150-bp library was prepared for each sample using the TruSeq DNA PCR-Free Library Prep Kit (Illumina), and sequenced on the Illumina HiSeq X platform. We expanded our initial 2-generation cohort by including sequences generated by a parallel project at the SNPRC on the Illumina HiSeq 2500 platform [58]. Thus, we analyzed a total of 29 baboons distributed into 2 pedigrees. The $P_0$ generation is composed of 3 dams and 2 sires that have 12 $F_1$ offspring between them. $F_2$ offspring are also present for a subset of these $F_1$ baboon individuals. Pedigree 1 also includes the parents of the $P_0$ individuals. We obtained an average coverage of 39X in the $F_1$ generation and 52X in the $P_0$ generation (S3 Table and S1 Data). We next set out to identify DNMs and estimate trio-specific error rates in a similar manner across both baboons and humans with the aim of minimizing the technical biases that can confound interspecies comparisons. We note, however, that systematic differences between the 2 Illumina platforms used for sequencing—although largely unavoidable—could potentially influence our final mutation rates.

### Calling SNPs and indels

We mapped our human (Hutterite) and baboon DNA sequences to the hs37d5 and papAnu2 reference genomes, respectively. To do so, we first interleaved the FASTQ formatted paired-end reads with seqtk mergepe (version 1.2-r101-dirty, github.com/lh3/seqtk) and trimmed sequencing adapters with trimadap (version r11, github.com/lh3/trimadap) [59,60]. Having thus prepared the sequences, we next used the Burrows Wheeler Aligner, bwa mem (version 0.7.9a-r786, bio-bwa.sourceforge.net), to align the human and baboon reads to the hs37d5 and papAnu2 reference genomes, respectively, and added read groups as needed [61]. We further processed the aligned sequences by marking duplicates using samblaster (version 0.1.24, github.com/GregoryFaust/samblaster) and then sorting the alignments with samtools sort (version 1.3, www.htslib.org) [62,63]. Finally, we used MergeSamFiles and MarkDuplicates from Picard Tools (version 2.9.0, broadinstitute.github.io/picard) to merge read groups by sample and perform a final round of duplicate marking, respectively, resulting in a BAM file of processed alignments for each individual [64]. Default parameters were used for all the aforementioned software, unless otherwise noted.

We further improved our read mappings by running them through GATK (version 3.7, gatk.broadinstitute.org) IndelRealigner and Base Quality Score Recalibration, in keeping with the GATK Best Practices workflow [41,65]. These 2 steps benefit from the inclusion of sets of confident SNPs and/or indels. For humans, we used dbSNP build 138 available from the

GATK resource bundle for the calibration. Genomic resources are more limited for *P. anubis*; there does not yet exist a widely agreed-upon compendium of common baboon SNP and indel variation. To overcome this deficit, we used 2 datasets to aid in calibration. We constructed the first dataset by making SNP and indel calls in our initial set of aligned reads using GATK's HaplotypeCaller and GenotypeGVCFs tools. (We ran HaplotypeCaller with the pcr_indel_-model parameter set to NONE to prevent the algorithm from incorporating PCR error-corrections on the variant likelihoods, because we sequenced our reads using a PCR-free protocol.) We then used bcftools isec (version1.3, www.htslib.org) to identify variants that were fully transmitted across all 4 generations of our baboon pedigree (i.e., variants in individuals 32043, 32849, and 33863 that were also observed in their respective parent, grandparents, and great-grandparents), reasoning that these are likely real polymorphisms [66]. Our second dataset consisted of the SNP and indel variants called by Robinson and colleagues in a sequencing panel of 100 baboons [58]. We excluded 14 baboons that overlapped with our pedigrees, leaving only variants called in the 86 unrelated baboons, which we subjected to a number of site-level "hard filters" as suggested by the GATK Best Practices workflow ("QualByDepth" > 2, "FisherStrand" < 60, "RMSMappingQuality" > 40, "MappingQualityRankSumTest" > −12.5, "ReadPosRankSumTest" > −8, "StrandOddsRatio" < 3 for SNPs) [41,58]. The aforementioned procedure thus yielded 1 set of "known" variants for humans (from dbSNP) and 2 sets for baboons (from transmission and an 86-baboon sequencing panel).

For each species, we supplied the appropriate sets of known variants along with the initial set of mapped reads to GATK's IndelRealigner and Base Quality Score Recalibration tools to produce reliable alignments for each individual. These alignments were in turn used to call SNP and indel variants using GATK HaplotypeCaller with the pcr_indel_model parameter set to NONE and the heterozygosity parameter set to 0.001 for humans and 0.002 for baboons, to account for their roughly 2-fold higher heterozygosity levels [58,67]. Finally, genotype likelihoods were jointly assigned across all samples of each species using GenotypeGVCFs and the species-appropriate heterozygosity, as in the previous step [68].

## Identifying DNMs

Having called variants in both humans and baboons, we next focused on identifying DNMs in the trios available in our 2 cohorts. Specifically, we searched for point mutations present in the offspring (heterozygous for the ALT allele) that were absent in both parents (homozygous for the REF allele), thus constituting a Mendelian violation. Using bcftools query, we identified 374,289 such sites across our 10 human trios and 797,900 sites across our 12 baboon trios. As most of these sites are likely false positives, we identified possible sources of errors and developed a number of filters to increase our specificity.

First, we applied a number of site-level filters to avoid poorly sequenced or misaligned regions of the genome. For each individual, we constructed upper and lower cutoffs on the depth of read coverage by performing a 2-sided Poisson test on depth at each site under the null hypothesis that the lambda ($\lambda$) parameter was equal to the mean read depth of that individual (as reported by the GATK DepthOfCoverage tool). We required that, for a given trio, a site must have a read depth that yields a *p*-value > $2 \times 10^{-4}$ in all 3 members of the trio to be considered callable. To this end, we identified the depth cutoffs of the rejection region and used these values as the minDepth and maxDepth arguments to GATK's CallableLoci tool, which outputs bed files delineating callable regions. We used the bedtools intersect tool to intersect these regions across the members of each trio [69]. In addition to filtering on depth, we required called SNPs to be biallelic and to have a quality (QUAL) score greater than 100. We also hard-filtered sites using the settings recommended by the GATK Best Practices

workflow ("QualByDepth" > 2, "FisherStrand" < 60, "RMSMappingQuality" > 40, "Mapping-QualityRankSumTest" > -12.5, "ReadPosRankSumTest" > -8, "StrandOddsRatio" < 3 for SNPs) [41]. We further removed known variants, filtering out SNPs if they were observed as variants in a panel of 87 Hutterite individuals (which did not include members of our sample) or the 86 baboon sample described in "Calling SNPs and indels" [58,70]. For a given trio, we also filtered out SNPs if the same ALT allele was observed in an unrelated family.

We reasoned that false positive DNMs might be introduced by miscalling truly homozygous sites as heterozygous in the $F_1$. To guard against this possibility, we required SNPs of a given offspring to have a genotype quality (GQ) greater than 40 and that the number of reads supporting the alternate (mutant) allele—i.e., the allelic depth (AD)—be greater than 3. We also performed, for each SNP, a 2-sided binomial test on allelic balance (AB), the proportion of reads supporting the alternate allele, under the null hypothesis that the frequency = 0.5, and discarded SNPs with $p$-values less than 0.05. We note that a frequency of 0.5 implicitly assumes that the mutation is constitutive in all cells of the offspring rather than mosaic.

False positive DNMs might also be introduced because of miscalling truly heterozygous sites as homozygous in the $P_0$. To remove such errors, for each trio, we required AD = 0 in at least one parent and GQ > 40 in both parents. We took the added step of removing DNMs determined to be in clusters, which we defined as groups of DNMs of size 3 or larger (see the section "Detecting DNM clusters" for our reasoning and details on cluster detection).

A comprehensive list of all of the applied filters is available in S4 Table. After filtering, we were left with 980 and 475 DNMs in the human and baboon cohorts, respectively. For trios in which a third-generation individual was also sequenced, we determined DNMs to have been "transmitted" if at least 2 reads supporting the DNM were present ($AD \geq 2$) in that individual.

### Estimating size of the accessible, orthologous genome

Obtaining a mutation rate per bp requires calculation of the total number of bps for which our approach provides high power to call mutations, the so-called accessible genome size. To this end, we defined the size of an individual's accessible haploid genome as the number of sites that were annotated as "CALLABLE" by the GATK CallableLoci tool and passed the known variation filters, as described previously. For a given trio, we further restricted the accessible haploid genome to the intersection of accessible regions across all 3 members of the trio and removed sites if at least 1 parent had a nonhomozygous reference genotype, resulting in a single bed file of accessible genomic regions for each trio. This procedure of subtracting and intersecting genomic regions was performed using the bedtools toolkit (version v2.23.0, github.com/arq5x/bedtools2) [69]. (The size of the accessible diploid genome is then simply twice that of the haploid genome.)

In order to make interspecies comparisons in later analyses, we further identified the segments of the baboon and human genomes that are not only accessible but also orthologous. To do so, we first identified orthologous genomic regions using the Enredo-Pecan-Ortheus (EPO) 7-primate multi-alignment dataset (ENSEMBL release 76) [71,72]. We used tools from the mafTools package (version 0.1, github.com/dentearl/mafTools) to preprocess the original MAF-formatted files by (1) removing duplicates using mafDuplicateFilter; (2) extracting human–baboon autosome multi-alignments using mafFilter; (3) coercing the multialignment block coordinates to use the positive (sense) Homo sapiens strand using mafStrander [73]. We then used a custom awk script to read the processed MAF files and create a set of bed files containing the human and baboon reference coordinates of each pair of orthologous regions. Separately, we intersected the accessible regions across all trios in each species cohort, resulting in an accessible haploid genome for each species. We then used the liftOver tool (www.genome.

ucsc.edu/cgi-bin/hgLiftOver) to convert the accessible human coordinates from the original hs37d5 reference to GRCh38, the reference genome used in the EPO multi-alignment [74]. Lastly, we performed a final round of intersections between each of the cohort-specific accessible genomes and our set of orthologous regions, resulting in a set of bed files defining a set of regions that were orthologous between the 2 species. The total size of the orthologous regions equaled 1,631,476,416 bp or about 64.6% and 69.7% of the mean callable genome of humans and baboons, respectively. Within the orthologous regions, we identified 505 human and 248 baboon DNMs (equivalent to 51.5% and 52.2% of DNMs originally called across all callable sites).

### Estimating the FNR by pedigree simulations

Our stringent filtering process likely removed true DNMs, which would lead us to underestimate the mutation rate. In order to measure the FNR of our DNM-calling pipeline, we simulated a set of "real" DNMs for each trio using information from other members of the pedigree and quantified the proportion of DNMs that were removed by our filters (S2A Fig). Specifically, we created these simulated DNMs by searching our initial set of SNP variant calls for sites satisfying the following criteria: (1) The $F_1$ and exactly 1 $P_0$ parent (hereafter referred to as the heterozygous parent) of a given trio were both heterozygous for the alternate allele (genotype 0/1); (2) the other $P_0$ parent was homozygous for the reference allele (genotype 0/0); (3) a $P_0$ parent (hereafter referred to as the substitute parent) of the same sex as the heterozygous parent but from a different family was homozygous for the reference allele (genotype 0/0); (4) the site passed hard filtering and depth filters. Of the sites that met these criteria, we randomly sampled 10,000 sites. We then transformed these Mendelian transmissions into non-Mendelian DNMs by treating the substitute parent as the real parent of the offspring in question, in effect "replacing" the genotype and quality statistics of the true heterozygous parent with those of the homozygous substitute parent. Where multiple individuals could be used as the substitute parent, we selected one at random for each DNM.

In practice, we performed this substitution by removing the heterozygous parent as well as its descendants save for the focal $F_1$ and $F_2$s and subsequently rerunning the joint genotyping algorithm (GATK GenotypeGVCFs) across the remaining individuals. Removing these related individuals and re-genotyping was necessary because transmitted heterozygous variants will be shared quite often with sibling $F_1$s—much more often than is the case for true DNMs [2]. Because the joint genotyping algorithm improves the genotype likelihood of an allele if it is observed in multiple individuals, the removal of other individuals that may carry the same allele mimics as closely as possible the status of a true DNM. The aforementioned procedure was performed twice for each trio, in one case replacing the mother and in the other case replacing the father, resulting in 2 sets of simulated DNMs for each trio.

Next, we applied the remaining filters to the set of simulated DNMs, as if they were observed ones. The proportion of simulated DNMs that either failed the filters or were not retained during the re-genotyping stage served as our FNR estimate for each trio.

### Estimating the FNR by DNM-insertion simulations

The reference genome quality differs between humans and baboons, which may lead to differences in the reliability of reads mapping. Errors in mapping could lead true DNMs to be missed if reads carrying DNM are not successfully mapped, or to the creation of spurious DNMs, if they are mapped incorrectly. To gain a sense of the FNR due to mapping errors, we used a DNM-insertion approach, in which we modified base calls in the reads themselves in order to create a synthetic point mutation. To this end, we used the addsnv.py tool from the Bamsurgeon

suite (github.com/adamewing/bamsurgeon), which inserts a given variant into a chosen pro-
portion of the reads at a predetermined site and realigns the region [75]. We applied the tool to
2 baboon $F_1$s (17199 and 19181) and one human $F_1$ (H3) individual. We also ran addsnv.py on
the $F_2$ offspring of the $F_1$ individuals, if one was present. We randomly targeted a total of 1,000
sites in $F_1$s without an $F_2$ and 2,000 sites in those with an $F_2$ (5,000 sites total across the 3 trios);
we restricted these sites to callable and nonvariable (i.e., monomorphic sites more than 10 bp
away from an indel) regions of the genome and further ignored sites where SNPs were observed
in the unrelated cohorts. To simulate transmission, for each site, we randomly chose to insert
the SNP into the $F_2$ with probability 0.5. For a given individual ($F_1$ or $F_2$), we chose the AB at
each targeted site by first determining the depth at the site using samtools depth and then ran-
domly sampling from an empirical distribution of the AB at heterozygous sites in which one $P_0$
parent is homozygous for the REF allele and the other is homozygous for the ALT allele. In
order to account for nucleotide context, we randomly selected the ALT allele at the target site by
sampling from distributions of the relative abundance of 96 possible trinucleotide mutation
types, which we compiled using DNM data from Jónsson and colleagues [17] for humans and
Robinson and colleagues [58] for baboons (see "Comparing mutation spectra").

Having thus designed our target DNMs, we ran addsnv.py on the finalized BAM alignments
(post–Base Quality Score Recalibration) of the selected individuals. We set the snvfrac parame-
ter to 0.1, the coverdiff parameter to 0.5, and the aligner parameter to "mem." To speed up
computation, we extracted reads in a 20-kb window centered on each target site and ran
addsnv.py in parallel across sites. Overall, the tool successfully inserted DNMs at 85% of the
target sites. We subsequently used GATK HaplotypeCaller and GenotypeGVCFs to call vari-
ants from the postinsertion BAM alignments in the same manner as in our variant-calling
pipeline described previously. We further passed the BAMs to GATK CallableLoci and ignored
sites if they failed to elicit a "CALLABLE" annotation, because such sites do not enter calcula-
tions of our mutation rate estimates. After this step, we were left with 4,112 synthetic DNMs,
or approximately 82% of the 5,000 initially targeted sites.

One drawback of the Bamsurgeon addsnv.py tool is that it does not insert variants into the
reads in either a phase- or LD-aware manner. Absent sequencing and alignment errors, reads
from a diploid individual should correspond to at most 2 haplotypes; inserting variants with-
out consideration for which reads already carry alternate alleles can generate 3 or more haplo-
types. GATK HaplotypeCaller builds a local haplotype graph as part of its algorithm and will
thus down-weight the likelihoods of synthetic variants that have been inserted without respect
for phase [41]. To guard against this issue, we developed a "3-haplotype" test that operates
analogously to the 4-gamete test [76]; a synthetic DNM fails the 3-haplotype test if, when pair-
ing it with another heterozygous SNP, 3 or more haplotypes can be detected in the reads. To
lessen the effects of sequencing and alignment error, when identifying SNPs for phase-com-
parison against the focal DNM, we only considered SNPs that had an AD of 3 or more for the
ALT allele and a $p$-value $> 0.05$ in a 2-sided binomial test on AB (null AB = 0.5). We addition-
ally required 2 or more reads to support a haplotype for the haplotype to be counted. Using a
custom script, we applied the 3-haplotype test to our pool of synthetic DNMs, of which
approximately 30% failed the test. The 2,908 synthetic DNMs that passed the 3-haplotype test
were subsequently subjected to our DNM-calling filters to obtain FNR estimates. As before,
the proportion of simulated DNMs that either failed the filters or were missed during variant
calling served as our FNR estimate for each trio.

In both species, the resulting Bamsurgeon FNR estimates were on average approximately
1.8-fold higher than what we had estimated using simulations starting with aligned reads. The
net effect on the mutation rate is unclear, as we also expect an increase in false discoveries due
to mapping, but those are harder to assess. Even ignoring false discoveries and taking the

Bamsurgeon results at face value, the ratio of yearly mutation rates in humans and baboons would be unchanged, at 33% (95% CI 15%–49%). The absolute mutation rates per generation would rise by approximately 8% in both species, from $1.23 \times 10^{-8}$ to $1.34 \times 10^{-8}$ per bp in humans and from $0.57 \times 10^{-8}$ to $0.62 \times 10^{-8}$ per bp in baboons. Given these results, it is likely that our pedigree-simulation strategy somewhat underestimates FNR; however, because this phenomenon appears to affect both species similarly, the impact on our overall conclusions is minimal.

## Phasing DNMs

Having called our DNMs, we subsequently identified the parental backgrounds on which they arose by phasing the genotypes. To this end, we used both a read-backed phasing method, which takes advantage of the phasing information contained in short-read sequences, as well as a phase-by-transmission approach, which was feasible because of the 3 generations present in our pedigrees.

For each species, we performed read-backed phasing by feeding the appropriate SNP variant call-set VCF file into the GATK ReadBackedPhasing tool with the phaseQualityThresh parameter set to 20, producing a new VCF file in which phased SNPs were annotated with haplotype identifiers. We assigned a given haplotype to a particular $P_0$ parental background if at least one SNP along the haplotype was unambiguously inherited from the parent in question. If more than one informative SNP was present in a haplotype, we required that all such SNPs indicate the same background; haplotypes that indicated conflicting parental backgrounds (i.e., if both maternally and paternally inherited SNPs were present) were given an ambiguous status. Using this procedure, we succeeded in identifying the parental backgrounds of 332 of the 980 human DNMs identified both within and outside of the orthologous regions. We successfully phased a higher proportion of baboon DNMs (238 of 475 total across the callable genome) as expected from their higher nucleotide diversity levels [67].

In order to phase DNMs by transmission, we employed a 2-step process of pedigree-based phasing using the SHAPEIT2 (v2.r904, mathgen.stats.ox.ac.uk/genetics_software/shapeit/shapeit.html) program and then inferred the parental background from transmission observations in the third generation [77,78]. We first converted our VCF-formatted SNP sets to BED/BIM/FAM format using PLINK-1.9 (www.cog-genomics.org/plink/1.9/) [79,80]. We also filtered away SNPs if they failed our hard filters, had poor GQ (<20), were multiallelic, or had excess missingness across the set of samples (>5%). Along with SNP observations, SHAPEIT requires a genetic map to weight the prior probabilities of transitioning between haplotype segments. For our human cohort, we used the HapMap Phase II b37 genetic map [81,82]. For baboons, we used the fine-scale recombination rate map generated by Robinson and colleagues [58], who applied LDhelmet under a range of "block penalty" values to 100 *P. anubis* genomes to generate estimates of the population recombination rate ($\rho$) between pairs of adjacent SNPs [83]. We converted the map created with a block penalty of 5 from units of $\rho$ per bp to centimorgans (cM) using a method by Booker, Ness, and Keightley [84]. Briefly, we weighted $\rho$ by the physical distance between adjacent SNPs and renormalized it such that the sum of weighted $\rho$ across each chromosome equaled the genetic map length of the chromosome (in cM/Mb) in *P. hamadryas* [85].

Having thus obtained a filtered SNP set and genetic map for each species, we fed these inputs into SHAPEIT, which we ran in the pedigree-aware duoHMM mode (flag --duohmm), with a mean window size of 2 Mb (flag -W 2). This step generated both maximum likelihood haplotypes as well as a haplotype graph for each cohort. To obtain crossover event probabilities, we next sampled 10 haplotype sets from the haplotype graph by rerunning SHAPEIT with different seeds, using the standalone DuoHMM program to generate recombination maps for

each sample, and averaging over the maps using mapavg.py [86]. Finally, we determined the phase of any given transmitted DNM by identifying the backgrounds of nearby in-phase SNPs in the third generation that were unambiguously inherited from the proband; we assigned the transmitted DNM the opposite background if a recombination event with probability >50% was present before the first informative SNP on both sides of the DNM site.

For each DNM, we combined phasing calls from the read-backed and pedigree-based methods into a single consensus call (paternal or maternal). Trivially, if only one phasing method succeeded then we used that inferred haplotype; similarly, if both phasing methods assigned a DNM to the same haplotype then we chose that haplotype as the consensus phase. Across both orthologous and nonorthologous regions, we successfully phased a total of 121 DNMs in humans using both methods, of which 119 agreed in the haplotype call. We eliminated the 2 inconsistent DNMs from consideration when estimating $\alpha$ and inferring sex-specific DNM counts, under the assumption they are errors (see next).

In baboons, 46 DNMs were assigned haplotypes across the orthologous and nonorthologous genome using both the read-backed and pedigree-based techniques. Of these DNMs, one was transmitted to the single available $F_2$ offspring and disagreed in the assigned haplotypes; this DNM was ignored in subsequent analyses. Baboon $F_1$ individuals with 2 $F_2$ offspring afforded 2 opportunities to infer phasing using transmission observations. We identified only 1 DNM (chr2:112,593,279 in $F_1$ individual 7267) that was transmitted to two $F_2$ offspring and pedigree-phased to both paternal and maternal haplotypes. To resolve this conflict, we manually assigned this DNM to the maternal haplotype, the phasing call indicated using the read-backed technique. Finally, 6 baboon DNMs exhibited "partial linkage" in that they were observed in exactly 1 of the 2 available $F_2$ offspring despite those individuals having the same parental haplotype at the focal site. This pattern could arise if the mutation occurred during the development of the $F_1$ (rather than being inherited from the $P_0$ gametes), and thus it would not be identified every time its corresponding haplotype was observed [2]. Read-backed phasing was also available in 3 of the 6 DNMs, and in 2 cases, the transmitted haplotype was inferred. We ignored the single discordant DNM and assigned the remaining 5 DNMs to the transmitted parental haplotype.

## Detecting DNM clusters

For each trio, we defined a DNM as belonging to a "cluster" of DNMs if it occurred fewer than 100 bp away from a neighboring DNM in genomic coordinate space; a DNM cluster is then simply a set of 2 or more DNMs whose nearest neighbor distances do not exceed 100 bp. Using this working definition, we identified 56 and 34 clusters in our human and baboon datasets, respectively (S1A Fig). We considered a DNM cluster to be transmitted if we observed all the DNMs within the cluster transmitting to the $F_2$ generation.

Across both species, we identified 23 clusters containing 3 or more DNMs in $F_1$ individuals with an available $F_2$ generation (27 clusters across all trios). If clusters behaved like singlet DNMs (i.e., having the same expected probability of transmitting in each trio; see "Estimating FDRs") then we would expect to observe approximately 13 of the 23 clusters in the $F_2$ generation. In contrast, none of these clusters were observed to be transmitted (S1B Fig; Fisher's exact test $p$-value = $2.2 \times 10^{-5}$). This finding agrees with the results from Sanger resequencing, as detailed in "Sanger sequencing". In both species, DNMs co-occurring in clusters of size two (hereafter referred to as doublets) were transmitted to the $F_2$ generation at a much lower rate than singlet DNMs. In particular, we observed a doublet in the H14 trio, which was both transmitted to its $F_2$ child, H25, and verified as "present" by Sanger resequencing (we failed to verify any other DNM clusters in H14; see S4C Fig).

Based on these observations from transmission analysis and Sanger verification, we reasoned that, within the context of our pipeline, clusters of different sizes harbor different FDRs.

As clusters of size 3 or larger appeared to have an extremely high FDR (consistent with 100%), we filtered these clusters out entirely from our DNM datasets, resulting in the removal of 75 human and 35 baboon DNMs. We subsequently proceeded to estimate FDRs separately for singlet and doublet DNMs in the 2 species.

## Estimating FDRs

We used the fraction of putative DNMs that transmit from $F_1$ individuals to their $F_2$ offspring to estimate, for each trio, the proportion of putative DNMs identified by our pipeline that are true positives—i.e., one minus the FDR. This approach relies on the fact that a spurious DNMs called in the $F_1$ is exceedingly unlikely to be seen in the $F_2$.

We first consider the case of singlet DNMs. Ignoring for now the fact that not all $F_1$ baboons have $F_2$ offspring, let $m_i \in \{1,2\}$ be the number of $F_2$ offspring present for the $i$th $F_1$. We assumed that each true positive DNM called in the $i$th trio has probabilities $q_{i,j}$ of being observed in the $j$th $F_2$ ($j \in \{1,\ldots,m_i\}$) and that spurious DNMs will never be observed in the $F_2$ generation. Treating the $F_2$ status (observed or not observed) of each DNM as independent, the total number of DNMs observed in the $j$th $F_2$ is thus binomially distributed. If only a single $F_2$ is present ($m_i = 1$), then the number of observed singlet DNMs, $Z_i^{(1)}$, in the $F_2$ is modeled as:

$$Z_i^{(1)} | W_i^{(1)} = w_i^{(1)} \sim \text{Binomial}(w_i^{(1)}, q_{i,1}),$$

where the superscript (1) denotes singlet DNMs and $W_i^{(1)}$ is the number of true positive singlet DNMs.

If 2 sibling $F_2$s are present ($m_i = 2$), then there are 4 outcomes (X) for the $F_2$ generation: (A) the putative DNM can be not observed in both $F_2$s, (B) observed in the first $F_2$ but not the second, (C) observed in the second $F_2$ but not the first, and (D) observed in both $F_2$s. If we assume that the statuses of a DNM in 2 sibling $F_2$s are independent, then the number of observed DNMs falling in each of the 4 categories can be modeled as:

$$Z_i^{(1)} | W_i^{(1)} = w_i^{(1)} \sim \text{Multinomial}(w_i^{(1)}, q_i),$$

$$Z_i^{(1)} = [Z_{i,\text{A}}^{(1)}, \ Z_{i,\text{B}}^{(1)}, Z_{i,\text{C}}^{(1)}, Z_{i,\text{D}}^{(1)}],$$

$$q_i = [(1 - q_{i,1})(1 - q_{i,2}), q_{i,1}(1 - q_{i,2}), (1 - q_{i,1})q_{i,2}, q_{i,1}q_{i,2}],$$

where $Z_{i,\text{X}}^{(1)}$ is the number of singlet DNMs observed for outcome X.

We made use of our simulated DNMs, which we described in the context of estimating FNR (see "Estimating the FNR by pedigree simulations"), in order to obtain the probability $q_{i,j}$ of observing a DNM called in the $i$th $F_1$ in its $j$th $F_2$. We took this approach rather than assuming $\frac{1}{2}$ both because we do not have 100% power to detect a variant transmitted to the $F_2$ (though in practice close to it for the accessible genome) and because GATK increases the likelihood weights of variants that are observed multiple times in a cohort, leading to deviation from the expected 50% probability of observing a called variant in the $F_2$ at stringent filtering thresholds. Thus, for each $F_1$-$F_2$ pair, we estimated the proportion of simulated, filtered DNMs that were observed in the $F_2$, denoted $\tilde{q}_{i,j}$.

Because the baboon pedigree includes loops (resulting from inbreeding between close relatives) for baboon trios with individual 1X2816 as the sire, we multiplied $\tilde{q}_{i,j}$ by a corrective factor of $\frac{0.5}{0.625} = 0.8$ to account for the fact that simulated DNMs may be transmitted either through the $F_1$ (as is typical) or through 1X2816 when it is the masked, heterozygous parent.

The denominator of 0.625 is the sum of the joint probabilities of a randomly chosen SNP being transmitted and deriving from the $P_0$ dam (0.25) or the $P_0$ sire (0.375). Thus, formally:

$$q_{i,j} = \begin{cases} 0.8 \, \tilde{q}_{i,j} & \text{if sire is 1X2816} \\ \tilde{q}_{i,j} & \text{otherwise} \end{cases}.$$

Using the parameterization outlined above, the likelihood of the observed singlet DNM data for trio $i$ is thus:

$$L_i^{(1)} = \Pr(Z_i^{(1)} = z_i^{(1)} | W_i^{(1)} = w_i^{(1)}) = \begin{cases} B(z_i^{(1)}; w_i^{(1)}, q_{i,1}) & \text{if } m_i = 1 \\ M(z_i^{(1)}; w_i^{(1)}, q_i) & \text{if } m_i = 2 \end{cases},$$

where $B$ and $M$ are the binomial and multinomial probability mass functions, respectively. Finally, we obtained a point estimate of the FDR of singlet DNMs for each trio using the maximum likelihood estimate of the number of true DNMs, $\hat{w}_i^{(1)}$:

$$\text{FDR}_i^{(1)} = 1 - \frac{\hat{w}_i^{(1)}}{y_i^{(1)}},$$

where $y_i^{(1)}$ is the total number of singlet DNMs observed in trio $i$. In searching for the maximum likelihood estimate of $w_i^{(1)}$, we constrain this quantity to be strictly less than or equal to the total number of singlet DNMs, $y_i^{(1)}$, and greater than or equal to the total number of singlet DNMs observed in the $F_2$ generation.

Seven of our baboon $F_1$s lack any $F_2$ offspring, which obviously precludes the use of transmission information for inferring an FDR for these individuals. For these individuals, we estimated FDRs by relying on the correlation we observed between the estimated FDR for individuals with an $F_2$ and the mean coverage in $F_1$s (S9 Fig). Specifically, we regressed the estimated FDR values on the $F_1$ average depth of sequencing coverage; we then used the fitted linear regression to obtain point estimates of the FDR of baboon $F_1$s without $F_2$ offspring from their average depth of coverage.

For doublet DNMs, we calculated a single FDR across all trios, rather than a per trio rate, which was necessary given the rarity of doublets. We used the same framework described above for estimating FDR from transmission information for singlets, but aggregating variables over all trios in each species:

$$L^{(2)} = Pr(Z^{(2)} = z^{(2)} | W^{(2)} = w^{(2)}) = B(z^{(2)}; w^{(2)}, \bar{q}),$$

$$\text{FDR}^{(2)} = 1 - \frac{w_{\text{mle}}^{(2)}}{\sum_i y_i^{(2)}}$$

where $Z^{(2)}$ is the number of doublet DNMs observed in the $F_2$ generation, $W^{(2)}$ is the number of true positive doublet DNMs, $y_i^{(2)}$ is the number of doublet DNMs observed in the $i^{\text{th}}$ $F_1$, and $\bar{q}$ is the arithmetic mean of $q_{i,j}$ taken over all $F_1$-$F_2$ pairs. All these operations were performed using the R statistical computing language.

## Sanger sequencing

In order to assess whether our FDRs were reliable, we used Sanger sequencing to validate the presence or absence of DNMs called in one trio. Specifically, we designed primer pairs (S1 Table) targeting all 89 DNMs called both inside and outside of the orthologous regions in H14

(S4 Fig). We also attempted to validate an additional 14 putative "cluster" DNMs in H14 that had been removed for being members of clusters sized three or larger. The clustering phenomenon allowed us to target multiple DNMs with a single primer pair such that we required only 87 primer pairs (designed using Primer3 [sourceforge.net/projects/primer3/]) to target the 103 total DNMs [87,88]. We evaluated primer specificity using NCBI PrimerBlast [89]. Using the high-fidelity Phusion polymerase (New England BioLabs E0553L), we attempted to PCR-amplify the 87 target sites in all 3 members of the H14 trio (H14, H11, H12). A subset of the primer pairs yielded nonspecific products, as indicated by multiple bands in gel electrophoresis. We performed band stab on these nonspecific products, selecting the appropriately sized band and performing a second round of PCR with fewer cycles. In total, we succeeded in amplifying 54 sites. We performed a PCR product cleanup step to remove primers and small DNA products using Agencourt AMPure XP Beads (Beckman Coulter B37419AB) or spin columns (Qiagen 28104). Finally, we submitted the cleaned products as well as their respective forward primers to Genewiz for Sanger sequencing.

We visually inspected the resulting chromatograms using the SnapGene Viewer (version 4.3.5, www.snapgene.com/snapgene-viewer/) and determined that Sanger resequencing succeeded in all 3 trio individuals for 25 of the 54 sites originally submitted [90]. These 25 sites contained a total of 7 cluster and 24 noncluster DNMs. We verified that all 7 cluster DNMs were absent in support of our decision to remove cluster DNMs from our dataset. In contrast, only 1 noncluster DNM was verified as absent (see S4B and S4C Fig for chromatogram examples). We compared the distributions of validated versus spurious mutation counts in our Sanger results (23 versus 1) against that inferred from transmission data. Of the 89 (noncluster) DNMs observed in H14, 44 were observed in its offspring, H25, out of an expected $89q_{H14,1} = 89 \times 0.492 \approx 44$, suggesting that there were no spurious DNMs. Comparing 1 out of 24 to 0 out of 44 by a Fisher's exact test yielded a 2-tailed $p$-value of 0.35 (S4A Fig).

## Calculating per-generation mutation rates and age effects

We inferred the number of "real" DNMs present in each trio offspring by adjusting the number of putative DNMs for FDRs, FNRs, and the size of the accessible genome. The numbers of singlet and doublet mutations observed in the $i$th $F_1$ are $y_{i,x}^{(1)}$ and $y_{i,x}^{(2)}$, respectively, with the subscript $x$ indicating the species. To obtain $v_{i,x}^s$, the error-adjusted number of mutations arising on a parental background of sex $s$ ($s \in \{female, male\}$) in species $x$ ($x \in \{human, baboon\}$), we used the relationship:

$$v_{i,x}^s = \text{int}\left[\frac{a_{i,x}^s(y_{i,x}^{(1)}(1 - \text{FDR}_{i,x}^{(1)}) + y_{i,x}^{(2)}(1 - \text{FDR}_x^{(2)}))}{(1 - \text{FNR}_{i,x})(O/H_x)}\right]$$

where $a_{i,x}^s$ is the proportion of mutations in the trio on background of sex $s$ ($a_{i,x}^m + a_{i,x}^f = 1$); $O$ is a constant denoting the amount of the genome that we determined to be orthologous and accessible (1,631,476,416 bp); $H_x$ is the haploid genome size of the species; and "int" is the nearest integer function. When an $F_2$ is present, we estimated $a_{i,x}^s$ to be the proportion of transmitted mutations that were phased to background $s$ (by both approaches or when no $F_2$ was available, using read-back phased mutations only). As described previously in Methods, we estimated $\text{FDR}_{i,x}^{(1)}$ and $\text{FDR}_x^{(2)}$ from $F_2$ transmission observations and $\text{FNR}_{i,x}$ from DNM simulations. We obtained values for $H_x$ by counting autosomal bps in the reference genomes of each species yielding $H_{hum} = 2,881,033,286$ bp and $H_{bab} = 2,581,196,250$ bp.

For each species, we model the number of de novo germline mutations inherited on a paternal (maternal) background as a function of the father's (mother's) age at conception, using a

Poisson distribution:

$$Y_x^s \sim \text{Poisson}(\beta_{x,0}^s + \beta_{x,1}^s(B_x^s - C_x)),$$

where $Y_x^s$ is the realized number of mutations; $B_x^s$ is the parental age at birth; $C_x$ is a constant denoting the typical gestation time of the species (S5 Table); and $\beta_{x,0}^s$ and $\beta_{x,1}^s$ are the sex-specific intercepts and parental age effects. We fit this regression model separately for each sex and for each species using our error-adjusted DNM counts $\upsilon_{i,x}^s$ and the recorded parental ages at birth (S2 and S3 Tables) of the $P_0$ individuals, and obtain the maximum likelihood estimates $(\hat{\beta}_{x,0}^s, \hat{\beta}_{x,1}^s)$ for the coefficients. Ages at birth were not available for baboon dams 1X0356 and 1X4519, and we therefore excluded their $F_1$ offspring from the baboon maternal regression analysis. Regression computations were implemented in R using the glm function.

To predict the per-bp mutation rate at some average generation time $G_x^s$ for each sex in each species, we considered:

$$\mu_{g,x}^s(G_x^s) = \frac{\hat{\beta}_{x,0}^s + \hat{\beta}_{x,1}^s G_x^s}{2H_x}.$$

By summing maternal and paternal rates, we obtained species-specific mutation rates per bp "per generation":

$$\mu_{g,x} = \mu_{g,x}^m(G_x^m) + \mu_{g,x}^f(G_x^f).$$

In the main text, we report mutation rate point estimates using paternal and maternal generation times thought to be typical of the 2 species.

For all mutation rates, we computed 95% CIs by generating 1,000 random draws of the regression coefficients $(\beta_{x,0}^s, \beta_{x,1}^s)$ from normal distributions centered at the point estimates and taking the standard errors of the fit as the standard deviations. Each replicate draw provided point estimates for the mutation rate of interest, from which we generated a distribution.

## Comparing mutation rates in repeat and nonrepeat regions

In order to assess the effect of repeat content on mutation rates in our 2 species, we categorized segments of each species' genome as repetitive or nonrepetitive based on the RepeatMasker and Tandem Repeats Finder reference genome annotation provided by the UCSC GenomeBrowser [74]. We merged this information with our orthologous regions dataset to split the callable genome of each individual of each species into 4 compartments: (1) nonorthologous and nonrepetitive, (2) nonorthologous and repetitive, (3) orthologous and nonrepetitive, and (4) orthologous and repetitive. For each species, we then calculated per-generation mutation rates in each of these compartments by collating data across all trios. We counted the number of mutations within a given compartment and weighted the contribution of each trio by its trio-specific FDR and FNR. Dividing the weighted sum by the total size of the compartment (summed across all trios) yielded a per-generation mutation rate for that compartment.

To test for mutation rate differences between pairs of compartments within a single species, we performed a Poisson rate ratio test. Specifically, we tested the null hypothesis that the mutation rate in compartment A is equal to that of compartment B ($H_0: \mu_A = \mu_B$) against the alternative that these rates were unequal. The test assumes that the number of DNMs arising in compartments A and B are independently Poisson-distributed with rate parameters

$$\lambda_A = N_A \mu_A,$$

$$\lambda_B = N_B \mu_B$$

where $N_A$ and $N_B$ are the total sizes of the 2 compartments in bp. We note that this test implicitly assumes that the underlying mutation rate of a compartment is the same across trios.

## Timing of life history events in baboons and humans

We estimated the length of various life history stages of humans and baboons using data drawn from the literature. For both species, we obtained values for the typical gestation time ($C_x$), male age at puberty, and male and female ages at reproduction ($G_x^s$), which are collated in S5 Table. Wherever we were unable to find data for *P. anubis*, we relied on estimates from other members of the *Papio* genus.

We drew typical gestation times from the AnAge database, using the value reported for *P. hamadryas* [91]. For typical male age at puberty in humans, we relied on an estimate of the median male age at spermarche by Nielsen and colleagues [35]; in baboons, we based our estimate on the median age at testicular enlargement measured in a wild population of *P. cynocephalus*, which maintains a hybrid zone with *P. anubis* in the Amboseli Basin of Kenya [33]. For typical male and female ages at reproduction, we used the mean paternal and maternal ages at conception of 1,548 Icelandic trios sampled in [17]. For baboons, we used estimates derived from field data on parental ages at birth from the same wild Amboseli population noted previously (Jenny Tung, personal communication).

## Quantifying germline cell divisions

Estimating the number of germ cell divisions per generation is challenging, as it requires accurate knowledge of cell division rates across the multiple developmental stages that span from formation of the zygote to production of the mature gametes. The major stages of this process are currently understood to extend (1) from the first post-zygotic division to sex differentiation, (2) from sex differentiation to birth, (3) from birth to puberty, and (4) from puberty to reproduction [16,38].

In humans, a classic reference is Drost and Lee, who estimated that stage 1 is comprised of 16 cell divisions [38]. They further estimated 21 cell divisions occurring during stage 2 in human males and 15 in females. Oocytogenesis is completed at the end of stage 2. In males, there are believed to be no cell divisions during stage 3. Under simplifying assumptions, the number of cell divisions in stage 4 depends on the length of the SSC cycle (16 days in humans) as well as the typical age at puberty (approximately 13 years) [37]. Assuming a typical male reproductive age of 32 years and taking into account the additional 4 cell divisions needed to complete spermatogenesis from spermatogonial stem cells yields $(32 - 13) \times \frac{365}{16} + 4 \approx 437$ cell divisions for stage 4 in male humans. Assuming these numbers, the male germ cell is thus the product of approximately 474 cell divisions postfertilization versus an average of 31 cell divisions in the female germline.

To the best of our knowledge, detailed analysis of germ cell division rates in baboons is limited to spermatogonial stem cells in stage 4, which are thought to divide every 11 days on average [34]. Assuming typical male ages at reproduction and puberty of 10.7 and 5.41 years, respectively, there are then about $(10.7 - 5.41) \times \frac{365}{11} + 4 \approx 180$ post-pubertal cell divisions in baboon males. For earlier developmental stages, we assumed that the number of cell divisions in stages 1–3 in the male baboon germline is the same as in humans, yielding a total of approximately 217 cell divisions. We discuss the impact of violations of these assumptions in the Discussion.

## Testing age effects and intercepts

We tested for differences between the parental age effects measured in our human and baboon datasets by performing, for each sex, a Poisson regression on the combined data from both species. This regression model included a slope (age effect) and intercept term, as before, as well as an interaction term for the effect of species on the intercept. After fitting this 3-parameter model with the glm function in R, we performed a LR test to compare it with the 4-parameter model previously obtained by regressing each species separately. We repeated the LR test on a Poisson regression model where we excluded transitions at CpG sites (67 in humans and 40 DNMs in baboons).

We also compared our estimated sex-specific mutation parameters to values obtained for the Jónsson and colleagues "deCODE" dataset by Gao and colleagues [17,32]. To this end, we relied on estimates made using only the 225 deCODE trios for which a third generation was available, a similar study design to ours. This choice was motivated by the observation that mutation rate estimates based on 3-generation pedigrees were different and likely somewhat more reliable than those based only on 2 generations [32]. Because Jónsson and colleagues reported $R$ = 2,682,890 bp as their callable genome, we multiplied the Gao and colleagues intercepts and age effects by a factor of $H_x/R$ to scale these values to the number of bp considered in this study (S1 Data). We then performed LR tests comparing our fitted models to these adjusted Gao and colleagues parameter values in R. We note that the Gao and colleagues parameters were estimated using a maximum likelihood framework to account for unphased mutations, whereas we use a normalization approach. The close similarity between our results, however, suggests that the effects of this difference are minor, and we therefore continued to use the Gao and colleagues parameters in further comparisons.

We tested whether our inferred sex-specific baboon DNM counts were consistent with a male-to-female mutation ratio, $\alpha$, that is 2.2-fold lower than that reported in humans. For the purposes of this test, we defined $\alpha$ as the ratio of paternal-to-maternal mutations at typical ages of reproduction (see section below for discussion on reproductive ages). Using the Gao and colleagues 3-generation age effect estimates yielded a human $\alpha$ value of 4.106 [32]. We thus formally performed a LR test on the null hypothesis that in baboons $\alpha$ = 4.106/2.2 = 1.87. We used the mle2 function from the bbmle R package (github.com/bbolker/bbmle) to compute maximum likelihood estimates (MLEs) using the BFGS optimization method and tested a range of initial values for $\alpha$ (under the free $\alpha$ model). We repeated the test for a 2.2-fold higher $\alpha$ in humans after removing transitions at CpG sites and rerunning the Poisson regression.

We also asked whether the mutation rate difference between human and baboon mothers was consistent with their typical ages at reproduction. To this end, we performed a LR test on the null hypothesis that the number of DNMs accumulated in human mothers at age 28.2 was 28.2/10.2≈2.76 times larger than the number of DNMs in baboon mothers at 10.2 years. We used the mle2 function to compute the MLEs using the BFGS optimization method and tested a range of starting values for the female mutation rate ratio.

In addition, we tested whether the age effect measured in baboon fathers was consistent with a model in which mutations track cell divisions—specifically one in which the paternal age effect in baboons, $\beta_{\mathrm{bab},1}^{\mathrm{m}}$, is 16/11 greater than that measured in humans, reflecting the ratio of their cell division rates. Using the human paternal age effect estimate from Gao and colleagues yielded the null hypothesis $\beta_{\mathrm{bab},1}^{\mathrm{m}} = 1.96$ [32]. We similarly tested whether the baboon paternal age effect was equal to that of humans (i.e., whether $\beta_{\mathrm{bab},1}^{\mathrm{m}} = 1.35$). For both tests, we calculated the MLE under the fixed $\beta_{bab,1}^{\mathrm{m}}$ model using mle2 and a range of starting values for the intercept, $\beta_{\mathrm{bab},0}^{\mathrm{m}}$. The MLE outputted by the glm R function in our initial age effect regression served as the likelihood estimate under the free $\beta_{\mathrm{bab},1}^{\mathrm{m}}$ model.

Finally, we examined whether humans and baboons exhibit differences in the number of DNMs accrued by puberty in males. To this end, we performed a Poisson regression of the estimated number of paternal DNMs in the 2 species on the number of years between puberty (taken to be age 13 in humans and 5.41 in baboons) and conception [33,35]. In this model, the intercept therefore corresponds to the onset of puberty. We fit a 2-parameter model with a slope and intercept term for both species, and a 3-parameter model including a slope and intercept term as well as an interaction term for the effect of species on the slope. To assess whether the number of DNMs accrued by puberty was significantly different between the species, we performed a LR test comparing the 3-parameter and 2-parameter models against a 4-parameter model that also allows the intercept to be different between species.

We additionally used the regression of paternal DNMs on years between puberty and conception to understand whether the difference in the number of DNMs accrued by puberty in human and baboon males reflects their respective ages at puberty. This amounted to testing whether the intercept is $13/5.41 \approx 2.4$ times higher in human than in baboon fathers. As before, we used the mle2 function to compute the MLEs using the BFGS optimization method and tested a range of starting values for the human-to-baboon ratio of male mutation counts at puberty.

### Testing an exponential maternal age effect model

Gao and colleagues showed an exponential age effect model to be a slightly superior fit to the maternal deCODE data than a linear model, a result apparently driven by mothers above the age of 40 [32]. Because mothers in our sample tend to be older than typical, we tested an exponential model as well. To this end, we calculated the AIC for our human maternal data under both the linear model of Gao and colleagues as well as their exponential model (using parameter estimates obtained from 3-generation families), applying the same genome scaling as before to both models.

### Estimating the male-to-female mutation ratio

In order to reliably compare the male-to-female mutation ratios, $\alpha$, between our species, we considered only those DNMs that were transmitted (i.e., observed at least once in an $F_2$), reasoning that this criterion should eliminate most false positive DNMs, which would otherwise bias our estimate downwards. We estimated $\alpha$ as the ratio of paternally- to maternally phased DNMs using the phasing results described in "Phasing DNMs" (see Fig 3A). We additionally computed estimates for $\alpha$ after removing transitions at CpG sites (CpG>TpG mutation types). For each species, we quantified the uncertainty in our estimate by segmenting the genome into blocks of a chosen size, bootstrap resampling these blocks 1,000 times, recalculating the male-to-female DNM ratio for each replicate, and then taking the 2.5%–97.5% quantile range across the replicates as our 95% CI. We used blocks of 50 cM width, based on the HapMap Phase II b37 genetic map [81,82] in humans and the Robinson and colleagues map in baboons (see "Phasing DNMs"). We performed a $\chi^2$ statistical test on the observed distribution of paternal and maternal DNM counts between the 2 species using the chisq.test function in R. We obtained similar estimates for $\alpha$ by using point estimates from our regression fits at the average ages of conception described above (Fig 3A).

### Comparing mammalian estimates of the male-to-female mutation ratio

We collected estimates of the male-to-female mutation ratio, $\alpha$, from published mammalian pedigree sequencing studies containing phased DNMs (Table 1, Fig 3B). When possible, we used the unaltered $\alpha$ estimates reported by the authors. If a ratio was not reported, we

calculated an $\alpha$ estimate using the ratio of the total number of DNMs phased to the male versus the female genome. For the Jónsson and colleagues [17] human estimate, we included only pedigree-phased DNMs from the 225 3-generation families. For each study, we also calculated the mean of the reported ages of the father at conception, excluding fathers if their $F_1$ offspring did not contribute phased DNMs to the $\alpha$ estimate. Because parental ages were unavailable in the cattle study, we relied instead on typical ages from the literature [26,27]. Note that since Besenbacher and colleagues reanalyzed the chimpanzee data generated by Venn and colleagues, one of the earliest nonhuman pedigree sequencing studies, we chose to reference only the results of Besenbacher and colleagues [21,22]. We performed a linear regression in R of the mammalian $\alpha$ estimates, including our own, against the mean paternal age of the study. To avoid duplicating species, estimates from Jónsson and colleagues (human) and Tatsumoto and colleagues (chimpanzee) were not included [17,23].

## Comparing mutation spectra

We examined the single-bp point mutation types of the transmitted DNMs, using the species' reference genome to categorize C>T transitions into those that occurred within (CpG>TpG) and outside of a CpG context (CpH>TpH), for a total of 7 mutation types. To obtain a baseline expectation for the "mutation spectrum" (i.e., the proportion of each mutation type), we compared our data to human DNMs published by Jónsson and colleagues as well as low-frequency polymorphisms in the sample of 86 unrelated baboons described previously [17,58]. We restricted our use of the Jónsson and colleagues dataset to 8,895 mutations found in the orthologous regions and called from 3-generation pedigrees, which is similar to our study design. For the unrelated baboons, we identified doubleton and tripleton SNPs within the orthologous regions that were genotyped in all 86 individuals. These low-frequency variants should reflect the broad patterns of the DNM spectrum, as most will have arisen recently and been only minimally affected by selection and biased gene conversion [30,31,92]. We inferred ancestral alleles and CpG context from the olive baboon multi-alignment against the rhesus macaque reference genome (rheMac2), using the liftOver tool to convert between coordinates [74]. SNPs were culled if the macaque allele did not match either allele (major or minor) or if one of the neighboring 5′ or 3′ positions was occupied by an "N" base, leaving a total of 2,650,098 SNPs. Uncertainty in the mutation type proportions was computed by bootstrap resampling 50 cM blocks 1,000 times and recomputing the proportions for each replicate. We used the 2.5%–97.5% quantile range across the replicates as our 95% CI.

To test for differences across datasets in mutation type proportions, we used the "forward variable selection procedure" implemented by Harris and Pritchard [6]. We first performed $\chi^2$ tests on all mutation types and then ranked the types by their $p$-values. We then iteratively removed the most significant mutation type and recalculated $p$-values for the remaining types.

## Comparing yearly mutation rates and substitution rates

To calculate yearly mutation rates per bp, $\mu_{y,x}$, we used the model of [42], which accounts for sex differences in generation times by dividing the sex-averaged mutation rate by the sex-averaged generation times:

$$\mu_{y,x} = \frac{2[\mu_{g,x}^{\mathrm{m}}(G_x^{\mathrm{m}}) + \mu_{g,x}^{\mathrm{f}}(G_x^{\mathrm{f}})]}{G_x^{\mathrm{m}} + G_x^{\mathrm{f}}}.$$

We note that here the generation times are the average over the lineage of species $x$ and thus comparable (in principle) to yearly neutral substitution rates. For humans, we assumed average paternal and maternal generation times of 32.0 and 28.2 years, respectively; in baboons, we

assumed an average generation time of 10.7 years in males and 10.2 years in females (see "Timing of life history events in baboons and humans" for description of age choices).

The number of neutral substitutions, $K_x$, arising in lineage $x$ over the $T_x$ years since the mean time to the human–baboon common ancestor is $K_x = \mu_{y,x} T_x$. Focusing on neutral sites, we therefore expect that

$$\frac{K_{\text{bab}}}{K_{\text{hum}}} = \frac{\mu_{y,\text{bab}} T_{\text{bab}}}{\mu_{y,\text{hum}} T_{\text{hum}}} = \frac{\mu_{y,\text{bab}}}{\mu_{y,\text{hum}}}.$$

We used the substitution rates at putatively neutral sites in human and baboon lineages reported by [29], for all mutation types. We used the randomly drawn regression coefficients described in "Calculating per-generation mutation rates and age effects" to propagate uncertainty in the sex-specific mutation rate estimates on to the relative per-year mutation rate (but treating FDR and FNR as fixed at their estimated values). We plotted the human and baboon lineages with a marmoset (*Callithrix jacchus*) outgroup using ggtree in R and with branch lengths corresponding to the substitution rates reported by [29] for all mutation types [93].

We were also interested in stratifying our substitution and mutation rate ratio comparisons by mutation type. We categorized our DNMs into the different possible mutation types involving strong (S: G/C) or weak (W: A/T) bp, separating strong-to-weak (S>W) mutations based on whether or not they occurred at CpG sites, for a total of 5 different types. For each species, we calculated mutation rates by aggregating information across all trios. Specifically, we collated the mutations and mutational opportunities of a given mutation type across $F_1$s, weighted by their trio-specific FDRs and FNRs, and divided the resulting value by the mean ages of reproduction of mothers and fathers, yielding a yearly mutation rate estimate. Baboon $F_1$ individuals 7267 and 8395 were excluded because of incomplete $P_0$ age information. To obtain 95% CIs, we resampled 50 cM blocks within each individual and calculated the 2.5%–97.5% quantile range.

## Inferring split times of humans and baboons

We were interested in estimating the split time of humans (an ape) and baboons (an OWM). For humans and OWMs, the split time and the mean time to the common ancestor $T_x$ is relatively close, with an estimated difference of 2–5 My [51]. The value $T_x$ can be estimated from substitution rates, given an estimate of the yearly mutation rate but relies on sex-specific generation times along each lineage, which are unknown [42]. We therefore explored the effects of life history trait values on inferred mean times to the common ancestor ($T_x = K_x/\mu_{y,x}$) using a range of plausible values. To this end, we used the mean age of first reproduction of 14 New World monkey species (3.0 years) as the lower bound on paternal generation times for both species [52]. For the upper bound on human paternal generation times, we used 32 years, the mean paternal age at conception in a sample of 1,548 Icelandic trios [17]; for baboons, we intentionally selected an upper bound of 15 years that is greater than the reported paternal generation times of modern baboon populations (10.7 years) in order to allow for the possibility of older than present-day generation times in the evolutionary history of the baboon lineage (see "Timing of life history events in baboons and humans" for a discussion of generation times). We also varied the paternal-to-maternal generation time ratio from 0.8 to 1.2 [42]. We estimated $T_x$ using the autosomal substitution rate estimates from [29] as described in the previous section along with the adjusted Gao and colleagues values for the intercept and age effect in the 2 species (described in "Testing Age Effects and Intercepts"). Because the paternal baboon data differed slightly from the Gao and colleagues values, we additionally estimated $T_x$ using our MLEs for the intercept and age effect in baboons (S8 Fig).

## Supporting information

**S1 Fig. Size distributions and transmission status of clusters of DNM calls.** (A) Distribution of DNM cluster sizes (defined as the number of DNM calls within 100 bp of each other) in humans (left, teal) and baboons (right, orange). The height of each bar denotes the number of DNM clusters of the given size, as labeled on the x-axis. (B) Proportion of DNM clusters observed transmitted to the $F_2$ generation, by size, in humans (left, teal) and baboons (right, orange). For each species, the horizontal red line denotes the mean proportion of DNMs expected to be observed in the $F_2$ generation if called in the $F_1$, as determined from simulations (see Methods). Vertical lines denote the 95% CI, as determined by bootstrap resampling clusters by 50 cM blocks [58,81,82]. Underlying data for this figure can be found in S2 Data. cM, centimorgan; DNM, de novo mutation. (TIF)

**S2 Fig. Estimates of false negative and false discovery rates of DNM calls.** FNRs (in A) and FDRs (in B) of the DNM-calling pipeline estimated for each trio across humans (top, teal) and baboons (bottom, orange). In all plots, sample identifiers of the focal $F_1$ are provided as labels on the x-axis. FNRs were estimated by applying the filtering pipeline to a set of simulated DNMs (20,000 per trio). For FNRs, 95% CIs from bootstrap resampling of mutations are narrow and hidden by the points. FDRs were inferred using a transmission-based approach in which the proportion of DNMs called in the $F_1$ that were also observed in the $F_2$ was compared against an expected proportion determined from simulations. Only the 5 baboon $F_1$ individuals with at least 1 sequenced $F_2$ offspring are depicted. Vertical lines denote 95% CIs from bootstrap resampling of 50 cM blocks [58,81,82]. See Methods for details on error rate estimation. Underlying data for this figure can be found in S2 Data. cM, centimorgan; DNM, de novo mutation; FDR, false discovery rate; FNR, false negative rate. (TIF)

**S3 Fig. Pedigrees with sample identifiers.** Reproduction of Fig 1 with unique sample identifiers for each individual in the human (top row, teal) and baboon (bottom row, orange) pedigrees. Sanger sequencing validation experiments focused on putative DNMs called in $F_1$ individual H14, highlighted in lavender. Baboon mothers 1X0356 and 1X4519, highlighted in gray, lacked birth dates and were thus excluded from age effect analyses. DNM, de novo mutation. (TIF)

**S4 Fig. Sanger sequencing validation results and chromatogram examples.** (A) Comparison of 2 methods for inferring the FDR of DNMs in human $F_1$ individual H14. In one, the FDR was inferred by Sanger resequencing of putative DNMs (purple) and in the second, by a transmission-based approach (green) in which the number of DNMs observed in the $F_2$ offspring of H14 was compared to the expectation determined by simulations (see main text). Vertical lines indicate the 95% CI of the point estimates. For the Sanger data, the binomial CI is indicated; for the transmission-based method, bootstrap resampling of 50 cM blocks [81,82] was used to obtain the interval. Examples of 4-color chromatograms from Sanger resequencing of a genuine G/C>A/T singlet DNM (in B) and a cluster of 3 spurious DNMs (in C). Chromatograms for the mother (top row) and father (middle row) of H14 (bottom row) are provided. Magenta arrows indicate the positions of putative DNMs originally called from Illumina sequence alignments. Absolute positions of the DNMs are also given at the top of each column using hs37d5 reference genome coordinates. Note the overlapping G (black) and A (green) peaks at the genuine DNM site in H14, indicating a heterozygous AG genotype, and the absence of a similar signal at the clustered calls. Underlying data for this figure can be found in S2 Data. cM, centimorgan; DNM, de novo mutation; FDR, false discovery rate. (TIF)

**S5 Fig. Consistency of mutation rates in orthologous and repetitive compartments of the genome.** Human (teal) and baboon (orange) mutation rates were calculated across various subsets (compartments) of the genome. (A) Estimated mutation rates within and outside of regions orthologous between baboons and humans. Circles denote rates calculated using mutations identified within orthologous regions between the 2 species, whereas squares represent estimates calculated using all callable regions of the genome. The 95% CI shown around the estimates was obtained by resampling from the Poisson regression (see Methods). (B) Estimated mutation rates in compartments of the genome defined by their orthologous and repetitive status. Mutations were assigned to 4 mutually exclusive compartments based on whether they arose within or outside of orthologous and/or repetitive regions of the callable genome. For each compartment, mutation rates per generation were estimated by combining information across trios (see Methods). Compartments are nonorthologous and nonrepetitive (empty circles), nonorthologous and repetitive (filled circles), orthologous and nonrepetitive (empty squares), and orthologous and repetitive (filled squares). Vertical lines represent 95% CIs calculated by bootstrap resampling of 50 cM windows within individuals followed by aggregation of rates across individuals [58,81,82]. Underlying data for this figure can be found in S2 Data. cM, centimorgan.
(TIF)

**S6 Fig. Mutation spectra of transmitted and untransmitted DNMs.** Spectra are shown for all DNMs (both transmitted and untransmitted to $F_2$s) called in humans (teal); all baboon DNMs (orange); DNMs called by Jónsson and colleagues [17] in a large set of 3-generation pedigrees (blue); and low-frequency SNPs (doubleton and tripleton alleles) identified in a sample of 86 unrelated baboons (red). As in Fig 2C, each point indicates the relative proportion of each of 7 mutation types, as indicated on the x-axis. Reverse complement mutation types were collapsed together into a single type, and transitions at cytosine sites were split into those that occurred inside (CpG>TpG) or outside (CpH>TpH) of a CpG context. For all datasets, only DNMs and SNPs located within genomic regions classified as orthologous between humans and baboons were included. Vertical lines denote 95% CIs from bootstrap resampling of 50 cM blocks. CIs for estimates from the 2 reference datasets (blue and red) are small and largely hidden by the points. In humans, the proportion of T>G mutations differs significantly from the dataset by Jónsson and colleagues (forward variable selection $p$-value = 0.019); no other mutation types differed significantly in their proportions from the values in the reference set in either species (Jónsson and colleagues and the SNP panel, respectively). We note that given the inclusion of all mutation calls in $F_1$s, regardless of whether they were transmitted to the $F_2$, we expect some spurious calls to have been included in our de novo sets, more so than in the stringent set of DNMs shown in Fig 2C. Underlying data for this figure can be found in S2 Data. cM, centimorgan; CpG, 5′-cytosine-phosphate-guanine-3′; DNM, de novo mutation; SNP, single-nucleotide polymorphism.
(TIF)

**S7 Fig. Dependence of the mutation count on sex and age in baboons and humans.** (A) Combined display of Fig 2A and Fig 2B showing human (teal elements) and baboon (orange elements) estimated DNM counts (y-axis) versus parental age at conception (x-axis) together in a single plot. Each $F_1$ individual is represented by both a circular and triangular point denoting the number of mutations arising on the paternal or maternal genomes, respectively. Maternal points for 2 baboons were omitted because of a lack of data on maternal age. Solid and dashed lines indicate the Poisson regression maximum likelihood fit for males and females in each species. Shaded regions denote the 95% CIs on the regression coefficients. We failed to

reject a model in which the paternal age effects of humans and baboons are the same (LR test *p*-value = 0.19). However, we find tentative evidence that the slope in baboons differs from the one estimated for humans based on a much larger data set by Gao and colleagues [32] (LR test *p*-value = 0.046). (B) Estimated DNM counts from fathers versus number of years between puberty and conception. Colors and elements are as in (A). Under a model in which paternal mutations track cell divisions post-puberty and the number of cell divisions and per-cell division mutation rates are the same in the 2 species, we would expect a higher baboon paternal age effect and similar intercept in the 2 species (see main text). For each species, the solid line denotes the best fit obtained from a Poisson regression of estimated paternal DNM count on paternal years between puberty and conception (x-axis), where puberty was assumed to occur at 13 years in humans and 5.41 years in baboons [33,35]. We do not reject a model in which the age effect is the same between species (LR test *p*-value = 0.19). A model in which human and baboon fathers accrue the same number of DNMs by puberty in males is somewhat unlikely (LR test *p*-value = 0.047) and a model in which both the paternal intercept and age effect parameters are the same in the 2 species even more so (LR test *p*-value = $3.9 \times 10^{-5}$). Underlying data for this figure can be found in S2 Data. DNM, de novo mutation; LR, Likelihood Ratio.
(TIF)

**S8 Fig. Predicted divergence times of humans and OWMs as a function of parental ages using estimated mutation parameters.** Divergence times were predicted using mutation and substitution rates measured in humans (teal) and baboons (orange), across a span of historical generation times. Human mutation rates were inferred using age effect parameters produced by Gao and colleagues [32], which were estimated from much more data (and are thus more precise). For this figure, as opposed to Fig 4D, baboon mutation rates were calculated using the parameters estimated in this study. Shaded areas cover the divergence times inferred across a span of plausible paternal generation times (x-axis) and paternal-to-maternal generation time ratios (ranging from 0.8 to 1.2, indicated in purple). The dashed gray line marks a plausible upper bound for the split time inferred from the fossil record [49,50]. As can be seen, the qualitative conclusion is the same as in Fig 4D. Underlying data for this figure can be found in S2 Data. OWM, Old World monkey.
(TIF)

**S9 Fig. Estimated FDRs against F1 depth of coverage, across baboon trios.** The FDR of DNM calls was inferred using a transmission-based approach in the 5 baboon $F_1$ individuals (orange points) for which an $F_2$ offspring was available. The black line indicates the best fit of a linear regression of these estimated FDRs against the mean depth of sequencing coverage in the same individuals. The gray-shaded region denotes the 95% CIs of the intercept and slope of the regression. The slope of the relationship is slightly negative (−0.0059) but not significantly so (*p*-value = 0.24). The regression fit was nonetheless used to predict FDRs in the 7 baboon $F_1$s that lacked an $F_2$ generation from their mean depth of coverage. Purple hash lines on the fitted line mark the predicted FDRs of these 7 baboon $F_1$ individuals. As expected from the insignificant slope, using the mean FDR across all individuals instead does not change the qualitative conclusions and yields a very similar estimate of the mutation rate in baboons: $5.70 \times 10^{-9}$ instead of $5.74 \times 10^{-9}$ per bp per generation. Underlying data for this figure can be found in S2 Data. bp, base pair; DNM, de novo mutation; FDR, false discovery rate.
(TIF)

**S1 Table. DNA primers and results of Sanger sequencing in H14.**
(XLSX)

**S2 Table. Age, sex, and pedigree relationships of sequenced humans.**
(XLSX)

**S3 Table. Age, sex, and pedigree relationships of sequenced baboons.**
(XLSX)

**S4 Table. List of de novo mutation filters.**
(XLSX)

**S5 Table. Life history traits of humans and baboons.**
(XLSX)

**S1 Data. Putative de novo mutations in humans and baboons; coverage, false negative rates, false discovery rates, and callable genome sizes of human and baboon trios; National Center for Biotechnology Information accession numbers.**
(XLSX)

**S2 Data. Underlying numerical data for Figs 2A, 2B, 2C, 3A, 3B, 4A, 4B, 4C, 4D, S1A, S1B, S2A, S2B, S4A, S5A, S5B, S6, S7A, S7B, S8 and S9.**
(XLSX)

## Acknowledgments

We thank Katherine Naughton for help preparing DNA samples for sequencing; Peter Andolfatto and the Andolfatto lab for help with the validation experiment; William Wentworth-Sheilds for help with archiving the DNA sequence data, Ipsita Agarwal, Guy Amster, Ziyue Gao, and Arbel Harpak for comments on the manuscript; and Peter Andolfatto, Guy Sella, and members of the Przeworski and Sella labs for useful discussions.

## Author Contributions

**Conceptualization:** Molly Przeworski.

**Data curation:** Felix L. Wu, Alva I. Strand, Carole Ober, Jeffrey D. Wall.

**Formal analysis:** Felix L. Wu, Alva I. Strand, Priya Moorjani.

**Funding acquisition:** Laura A. Cox, Carole Ober, Jeffrey D. Wall, Priya Moorjani, Molly Przeworski.

**Investigation:** Felix L. Wu, Alva I. Strand, Priya Moorjani, Molly Przeworski.

**Methodology:** Priya Moorjani, Molly Przeworski.

**Project administration:** Molly Przeworski.

**Resources:** Laura A. Cox, Carole Ober, Jeffrey D. Wall, Molly Przeworski.

**Supervision:** Priya Moorjani, Molly Przeworski.

**Validation:** Felix L. Wu.

**Visualization:** Felix L. Wu.

**Writing – original draft:** Felix L. Wu, Molly Przeworski.

**Writing – review & editing:** Felix L. Wu, Alva I. Strand, Laura A. Cox, Carole Ober, Jeffrey D. Wall, Priya Moorjani, Molly Przeworski.

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
