## [Editor Report · Decision Letter 0]

20 Nov 2019

Dear Dr Wu, 

Thank you for submitting your manuscript entitled "A comparison of humans and baboons suggests germline mutation rates do not track cell divisions" for consideration as a Research Article by PLOS Biology.

Your manuscript has now been evaluated by the PLOS Biology editorial staff, as well as by an academic editor with relevant expertise, and I'm writing to let you know that we would like to send your submission out for external peer review.

IMPORTANT: I note that your paper has been submitted as a Research Article; however, we think that it would be better considered as a Short Report, and this is a possibility that I'd mentioned in passing to Molly a few months ago. So when uploading your additional metadata (see next paragraph), please can you select "Short Reports" as the article type? No re-formatting is required.

Please re-submit your manuscript within two working days, i.e. by Nov 22 2019 11:59PM.

Kind regards,

Roli Roberts

Senior Editor

PLOS Biology

---

## [Decision Letter · Decision Letter 1]

17 Dec 2019

Dear Dr Wu,

Thank you very much for submitting your manuscript "A comparison of humans and baboons suggests germline mutation rates do not track cell divisions" for consideration as a Short Reports at PLOS Biology. Your manuscript has been evaluated by the PLOS Biology editors, an Academic Editor with relevant expertise, and by three independent reviewers.

You'll see that reviewers #1 and #3 are broadly positive about your study, but have various requests for improvement – rev #1’s are mostly presentational, but rev #3 requests several additional analyses. By contrast, rev #2 identifies a number of potential flaws, raising the possibility that three crucial assumptions underlying your study might be violated. Rev #2 also queries some disparities with respect to several prior studies, and has a significant list of technical issues.

In light of the reviews (below), we will not be able to accept the current version of the manuscript, but we would welcome re-submission of a much-revised version that takes into account the reviewers' comments. We cannot make any decision about publication until we have seen the revised manuscript and your response to the reviewers' comments. Your revised manuscript is also likely to be sent for further evaluation by the reviewers.

We expect to receive your revised manuscript within 2 months. 

**IMPORTANT - SUBMITTING YOUR REVISION**

*NOTE: In your point by point response to to the reviewers, please provide the full context of each review. Do not selectively quote paragraphs or sentences to reply to. The entire set of reviewer comments should be present in full and each specific point should be responded to individually, point by point.

*Re-submission Checklist*

*Published Peer Review*

*PLOS Data Policy*

*Blot and Gel Data Policy*

Sincerely,

Roli Roberts

Senior Editor

PLOS Biology

REVIEWERS' COMMENTS:

Reviewer #1:

This study is interesting and adds important evidence to the ongoing puzzle of reconciling information obtained from two very different sources to the same underlying question of how mutations are generated in different species and between males and females within a species. 

The major observations have been: Paternally derived mutations outnumber maternally derived mutations ~3:1 when comparing trios of mother-father-child (obs 1). On evolutionary time scales, the substitution rates present (at least) two puzzles: firstly, species which have a set of correlated traits (lower generation time, lower body mass, higher metabolic rate,...) accumulate substitutions faster (obs 2); secondly, the substitution rate is lower in more recent branches of the primate evolutionary tree than in older branches (obs 3). 

A dogma in the field has been that mutations are mainly replicative in origin and that both obs 1 & 2 can be explained by the number of cell divisions leading to a germ cell.  

The authors argued very convincingly against this dogma firstly in Gao et al, and now add weight, also convincingly, to that with the addition of data from baboons. 

The authors also argue here that the idea of cell divisions cannot explain the differences between male rates in the two species (also convincing; a bit weaker in male/male comparison but strengthened by comparison between males and females within species) 

The authors make a strong case that mutations are indeed clock-like in the two species considered; with a higher clock rate in males than in females; and maybe a higher rate per year in baboons than in humans (not totally clear on this point; please see comments below) 

They also helpfully put some of the results in the wider context of other primates. It would be helpful if the data in Table 1 could be annotated with information about their relevant lineage (I had to look up whether the Owl monkey was OWM or NWM but perhaps that's just me) 

The work is sound and convincing. This is a strong paper, and my main criticism is easily addressable by a modest rewrite. I found it difficult to put the findings in the overall context of other pertinent observations regarding mutations and substitution rates in primates and the space of models that can explain the findings. The discussion is, not unreasonably, focussed on the number-of-cell-divisions hypothesis, which they have convincingly debunked. However, I think it would be good for the paper and for the field if the discussion be more forward looking. Specifically, to present how these data sit in the context of other (potentially conflicting or unexplained) findings. Specifically, does the primate data overall now (various primate pedigrees now available and reported in Table 1) suggest a stable mutation rate per year across primate species, rather than a dependence on generation time? In other words,to what extent is the generation time correlated with mutation rate per generation? How does the previously reported more-clock-like behaviour of CpG and and less clock-like behaviour of non-CpG mutations sit with these findings. While there appears to be no model that currently explains everything, it would be helpful to get a clear sense of which models have been down-weighted and which ones up-weighted by these new data for all of observations 1-3 above (and any other pertinent observations that I haven't listed). The authors highlight the conflict between the absolute branch lengths from substitution data and the DNM rate but is there also a conflict in the relative branch lengths?  

Detailed comments

1) In the introduction, various possibilities are mentioned to explain the generation time effect (pg 3, lines 70-79). it would be helpful to present the alternative explanations in the field as well (metabolic rate etc) 

2) pg 5, line 127. it would be worth unpacking and explaining the sentence "there should a stronger paternal age effect in baboons". there are two distinct elements to this -- both of which the authors later explore (within males comparison and male/female comparison) -- and would be helpful to explain that here. 

3) I found the paragraph pg 7 lines 178-192 very difficult to understand. Please clarify. I also found it difficult to see how (in the methods) 4% FDR became 0% FDR. At any rate the sample used in sanger sequencing is not representative of the FDR per se (which is much higher in most samples) but of the approach to estimate FDR, so the FDR of this sample should not be emphasized. 

4) pg 7, lines 204, 205, please give units of age effect (DNMs per year?) 

5) It would be helpful to have a supplementary figure in which the data from Fig 2A and 2B are presented in the same plot (the difference in between the X and Y axes makes it harder to compare the male human/male baboon rates with female human/female baboon rates). for example, it would be helpful to see if the baboon data extrapolates into human data 

6) pg 8 line 230. typo? the baboon rate is not 44% lower, it is 0.44*human rate i.e 56% lower 

7) pg 9, lines 237-239. please also show full analysis possibly in supplementary, not just with the subset of DNMs in F2 

8) pg 10. line 298. typo. "humans" twice 

9) pg 10 lines 271. is the data compatible with clock-like mutation in females i.e. is a 2.8 fold difference in the mutation rates compatible with the data 

10) pg 12 line 333. point 4. OWM mutation rates are about half that of apes: please clarify you mean per generation mutation rate 

11) pg 12. lines 343-355. this is an important paragraph but very difficult to understand. firstly the authors state that the damage rates must be similar across mammals and the data in this paper for the two species supports that. It would be helpful to add a column in Table 1 with an estimated per year mutation rate. however, a few lines later the authors say that the "per year mutation rates vary across species and are higher in short-lived ones". this seems to be a contradictory statement. I am guessing the authors mean substitution rates in more distant species. please clarify.  

12) discussion final paragraph -- strong arguments that the discrepancy between the evolutionary data and present day DNM estimates cannot be explained by changes in generation times alone. Can they be explained by changes in the mutational rates per se (e.g. due to changes in the environment)? If so, please mention in discussion

Reviewer #2:

Adding to the growing literature of direct mutation rate estimates in primates, Wu et al. estimate sex-specific mutation rates in baboons (Papio anubis) using resequencing data from two multi-generational pedigrees (29 individuals). Mutations either originate from errors occurring during DNA replication or from DNA damage during normal cell growth - however, the relative contribution of these mechanisms to the overall mutation rate remains unknown. As mutation rate is a quantitative trait, differences in life history traits (e.g., in physiological factors such as age) can lead to differences in rates between species. Extending their previous work in humans (Gao et al. 2016), the authors here investigate how mutations accumulate with age and cell division rates in baboons - an Old World monkey, closely related to humans, but with differences with regards to spermatogonial stem cell (SSC) division rate and mean age of reproduction. Specifically, the authors investigate whether human and baboon mutation rates reflect differences in life history, under a model in which mutations are assumed to be proportional to cell divisions. Assuming the number of cell divisions in the germlines to be 474 in male humans, 217 in male baboons, and 31 in females (both humans and baboons), a higher male mutation bias in humans as well as a stronger paternal age effect in baboons can be expected. However, neither is observed - either, as noted by the authors, because some of their underlying assumptions are violated, or, alternatively, because a subset of germline mutations might be due to damage rather than being replicative in origin. 

There are three assumptions that are critical to the study - and any violations might significantly change the expectations and interpretations of the results obtain in this study:

i) all germline mutations are replicative in origin, 

ii) human and baboon ova are the product of the same number of cell divisions, and 

iii) SSC division rates are 23 cell divisions per year in humans and 33 in baboons.

The assumptions raise several questions:

i) Assuming germline mutations to be replicative in origin, shouldn't the authors exclude CpG mutations from their study which are due to spontaneous deamination (i.e., non-replicative)? 

ii) Why would the same number of cell divisions be expected in human and baboon ova? And, more importantly, what are the expectations if this assumption is violated? 

iii) As noted in the Methods section, our current knowledge of germ cell division rates in baboons is limited to spermatogonial stem cells in stage 4. Given the large difference in rates observed between humans and baboons at this stage, why do the authors assume the number of cell divisions in stages 1-3 in the male baboon germline to be the same than in humans? 

Moreover, there are two results that warrant more detailed explanation/discussion:

i) This study did not detect any significant inter-species differences in the mutational spectra between humans and baboons. How can this observation be explained in light of Harris 2015 and Harris and Pritchard 2017 who identified both population- and species-specific mutational signatures among humans and between great apes? Additionally, in contrast to Harris 2015 and Harris and Pritchard 2017, SNPs were categorized assuming the major allele to be ancestral rather than inferring the ancestral allele using the primate multi-species alignment (which was used by the authors to identify orthologous regions).

ii) Assuming a molecular clock, the mutation rates reported here lead to a divergence time estimate of 63-67 million years between apes and baboons - much older than the 35 million years expected from the fossil record. Moreover, modelling changes in life history traits suggest that low generation times (3-5 years) would be required in both species to reconcile the fossil record with the estimates presented in this study. As noted by the authors, this seems rather implausibly given that "male baboons typically enter puberty around the age of six and reproduce four years later". Following Scally and Durbin (2012), the authors thus suggest a slowdown of mutation rate along the baboon lineage in order to reconcile these estimates. More discussion is needed here (indeed, it feels like the manuscript is ending rather abruptly), as the original Scally and Durbin (2012) paper argued that a hominoid slowdown, which is consistent with an increase in generation time, explains why humans have accumulated 30% fewer mutations than baboons. How would an additional slowdown in the baboon lineage fit into this? How likely is it that mutation rates have slowed in hominoids and baboons but not in other primates? Which (potentially shared) mechanisms could even be driving these slowdowns? 

In addition to the above, there are several methodological details that warrant attention as they are likely to impact the results of this study:

i) Variant Calling: Perhaps most importantly, the authors employed a multi-sample genotype calling strategy. Han et al. (2014) have shown that multi-sample genotype calling strategies lead to an under-calling of rare variants. As DNM are by definition rare, the authors are likely to miss genuine mutations, thus underestimating rates. Unfortunately, this issue becomes worse the more data is available, thus a stronger under-calling in baboons is expected compared to humans. As this can potentially lead to severe biases in the mutation rate estimates, I would strongly encourage the authors to confirm their results using a single-sample calling strategy. Moreover, the authors use the GATK's HaplotypeCaller with a default heterozygosity rate of 0.001, despite known higher diversity in baboons. 

ii) Estimating FNRs: As noted by the authors, false negatives as well as false positives can have strong effects on the estimated fraction of DNMS that are of paternal/maternal origin. Yet, the way DNMs are simulated to estimate FNR does not account for false negatives due to read mapping errors. Consequently, the reported FNR is likely an underestimate. Read processing and mapping are essential parts of the computational pipeline - with inevitably errors (e.g., due to reads mapping to locations different from the one they originated from or due to an inability to map reads against the particular reference genome). In order to incorporate false negatives due to issues with the downstream bioinformatic analyses (i.e., pre-variant calling), DNM should be simulated and put through the exact same computational pipeline than the actual data (e.g., Keightley et al. 2014). Moreover, with regards to the read alignment, repeat regions are well-known for causing mapping issues. As repeats have not been excluded in this study, could the authors please comment on how many DNM were observed in repeat vs non-repeat regions of the genome?

iii) Identification of DNMs: The authors require DNMs to be supported by at least 3 alternate reads in an offspring. DNMs are then classified as "transmitted" to the F2 if at least 2 reads support the DNM. It is unclear why the same read thresholds are being used, given the vastly different read coverage in the species (i.e., 24X in humans vs. 45X in baboons on average). To allow for fairer comparisons, the same percentage (rather than count) of alternate reads should be utilized.

iv) Male-to-Female Mutation Ratios: The authors state a lack of a fine-scale genetic map for baboons when estimating male-to-female mutation ratios. However, a fine-scale LD-based map is available from Robinson et al. 2019 and should indeed be used rather than assuming an (unrealistic) uniform map. 

v) Base Quality Score Recalibration: The authors note that there is no "gold standard" dataset available for Base Quality Score Recalibration in baboons. Robinson et al. 2019 published a variant dataset containing >50 million SNPs in 100 baboons that could be utilized here. Perhaps this is the second dataset from the sequencing panel of 89 unrelated baboons that the authors are referring to in their Methods section (please note that the cited paper, reference 49, is pointing to a study in cattle)? If it indeed is the same dataset, what are the differences in filtering strategy applied by the original work and this study? 

vi) The authors state that the application of different computational pipelines hinders the comparison of mutation rate estimates among primates. While I completely agree with this statement, the differences in the experimental design might also introduce biases. In this study, 100bp paired-end Illumina HiSeq 2000 sequencing was used for human individuals while 150bp paired-end Illumina HiSeq X sequencing was used for baboon individuals. There are two points of importance: i) Heng Li (and others) have previously noted that empirical error rates are higher in HiSeq X than in HiSeq 2000 and ii) longer reads usually align more reliably than shorter reads. Neither has been taken into account in the presented analyses (or, potential implications discussed). In addition, mean coverage differs markedly between the species (i.e., 24X in humans vs. 45X in baboons).

vii) Data Access: No code has been shared with the reviewers hindering a more thorough evaluation of the results.

Lastly, some minor comments: 

* Sex-averaged germline mutation rate estimates for humans range from 1.25 x 10-8 to 1.34 x 10-8 per base pair per generation in orthologous and all callable regions, respectively. A similar pattern is seen in baboons. Is there any explanation for the lower mutation rate estimates when only orthologous regions are considered?

* Estimates from Tatsumoto et al (2017) for chimpanzees were excluded from the comparative analysis. As Tatsumoto et al's estimates differ from those reported by Besenbacher et al. (2019), it would be interesting to check whether both agree with regards to the analyses presented here.

* line 533 states that 26 baboons were used in this study while other parts of the manuscript state 29 individuals.

Reviewer #3:

This paper presents an interesting and carefully conducted study of de novo mutation rates measured in humans and baboons. The authors find that baboon mutation rates per generation are approximately half that of humans, largely because of their lower generation time, and that consistency with their phylogenetic divergence seems to require a mutational slowdown in not just humans both both species. The ratio of maternal to paternal mutations is nearly the same in both species and inconsistent with the hypothesis of a replicative origin of most de novo mutations, adding additional evidence to a hypothesis that the authors previously published in an analysis of human data. Although I have a few suggestions for improving the manuscript, they are quite minor and are not make-or-break issues.

1. The authors specify that the mutation rates of the aligned callable genomic regions are slightly lower than the mutation rates of the entire callable genomic regions in each species. Although the magnitude of the difference is low, it is potentially significant if it means that the mutation rate of the currently callable genome might not be representative of the rest of the genome. Can the authors specify whether the mutation rates of the homologous regions and nonhomologous regions are significantly different, and whether any significant difference can be explained by differences in repeat content? Is mutation rate symmetrically elevated in the nonhomologous regions from both species? This could be important if it lends credence to the idea that human and baboon DNA might have mutated faster at an earlier point when it perhaps had higher repeat content or was otherwise functionally different than it is now. 

2. Unless I missed it, the manuscript doesn't test whether there's a significant difference between humans and baboons in the number of additional mutations accumulated per year of parental age or the y-intercept expected number of mutations already accumulated by puberty. It would be interesting to specify these to give readers a sense of which of these values appear static vs evolutionarily labile over these few million years of divergence. 

3. From Figure 1, it looks like paternal age might explain less of the variation in mutation rate between baboon families than between human families. Is this an accurate assessment, or is the excess dispersion of the baboon data points simply the result of lower mutation counts and higher noise?

Additional line by line comments:

Line 119 states that baboons enter puberty "around age 6 and reproduce four years later." Does this mean that age 10 is the mean age of first reproduction, or the mean age of a randomly sampled baboon's parent?

Line 280 states that "paternal mutations are accruing no faster with age in baboons than humans (LR test p-value = 0.045)." Doesn't the fact that this p-value is less than 0.05 imply a marginally significant difference, not a lack of difference?

---

## [Decision Letter · Decision Letter 2]

7 Jul 2020

Dear Dr Wu,

Thank you for submitting your revised Short Report entitled "A comparison of humans and baboons suggests germline mutation rates do not track cell divisions" for publication in PLOS Biology. I'ave now obtained advice from the original reviewers and have discussed their comments with the Academic Editor. 

Based on the reviews, we will probably accept this manuscript for publication, assuming that you will modify the manuscript to address the remaining points raised by reviewer #2. Please also make sure to address my Data Policy requests noted at the end of this email.

We expect to receive your revised manuscript within two weeks. Your revisions should address the specific points made by each reviewer. In addition to the remaining revisions and before we will be able to formally accept your manuscript and consider it "in press", we also need to ensure that your article conforms to our guidelines. A member of our team will be in touch shortly with a set of requests. As we can't proceed until these requirements are met, your swift response will help prevent delays to publication.

*Copyediting*

*Published Peer Review History*

*Early Version*

*Submitting Your Revision*

Sincerely,

Roli Roberts

Senior Editor

PLOS Biology

ETHICS STATEMENT:

-- Please include the full name of the IACUC/ethics committee that reviewed and approved the animal care and use protocol/permit/project license. Please also include an approval number.

-- Please include the specific national or international regulations/guidelines to which your animal care and use protocol adhered. Please note that institutional or accreditation organization guidelines (such as AAALAC) do not meet this requirement.

-- Please include information about the form of consent (written/oral) given for research involving human participants. All research involving human participants must have been approved by the authors' Institutional Review Board (IRB) or an equivalent committee, and all clinical investigation must have been conducted according to the principles expressed in the Declaration of Helsinki.

DATA POLICY:

Many thanks for depositing your raw data in dbGaP, SRA, and the zipped supplementary folder. In addition we ask that all individual quantitative observations that underlie the data summarized in the figures and results of your paper be made available in one of the following forms:

Regardless of the method selected, please ensure that you provide the individual numerical values that underlie the summary data displayed in the following figure panels as they are essential for readers to assess your analysis and to reproduce it: Figs 2ABC, 3AB, 4BC, S1ABCD, S2AB, S4A, S5AB, S6, S7AB, S9. It may be that some of these data are in your supplementary data file, but the relationship to the individual figure panels is unclear. NOTE: the numerical data provided should include all replicates AND the way in which the plotted mean and errors were derived (it should not present only the mean/average values).

REVIEWERS' COMMENTS:

Reviewer #1:

I am satisfied with the revisions and have no further comments or queries. 

Reviewer #2:

The authors have addressed the majority of my previous concerns in the revised version of their manuscript though please note that reviewers are still unable to access the code to evaluate the results as the GitHub link results in a 404 error (see my previous comment regarding Data Access). Overall, this paper adds nicely to the growing literature of DNM estimation in primates.

I only have a few minor comments on the revised version of the manuscript:

The authors mention several DNM studies in the main text that are absent from Table 1 (e.g., a direct mutation rate estimate for chimpanzees (Venn et al. 2014), green monkeys (Pfeifer 2017), and wolves (Koch et al. 2019)). To allow the reader to navigate and compare the current (and still limited) literature on the topic more easily, these should be added to Table 1. In addition, Table 1 in the response to the reviewer comments has two additional remarks (D: Rate estimate as reported in study and E: Data from marmoset, a New World Monkey.) that are missing from Table 1 in the manuscript. 

Early studies in humans were unable to detect the more subtle maternal age effect and only with subsequent sequencing of hundreds to thousands of trios such an effect was detected in the species. In this regard, it is remarkable that the authors were able to discern a maternal age effect in baboons with merely two pedigrees/29 individuals. Could the authors please comment on why, in their opinion, no maternal age effect has been reported in similarly powered studies of other non-human primates (e.g., Thomas et al. 2018; Besenbacher et al. 2019)? 

lines 272 and 273: "To understand how the distribution of single base pair mutation types (i.e., the mutation "spectra") (…)" 

It should be "distribution and mutation spectrum" (singular) or "distributions and mutation spectra" (plural).

Reviewer #3:

In my opinion, the authors have responded satisfactorily to all comments.

---

## [Editor Report · Decision Letter 3]

28 Jul 2020

Dear Dr Wu,

On behalf of my colleagues and the Academic Editor, Nick H Barton, I am pleased to inform you that we will be delighted to publish your Short Reports in PLOS Biology. 

Early Version

PRESS 

Kind regards,

Alice Musson

Publishing Editor, 

PLOS Biology

on behalf of

Roland Roberts,

Senior Editor

PLOS Biology